

# A hybrid Kolmogorov-Arnold networks-based model with attention for predicting Arctic River streamflow

Renjie Zhou [1,*], Shiqi Liu[2]

[1]: Department of Environmental and Geosciences, Sam Houston State University, Huntsville, TX 77340, USA;
Email: renjie.zhou@shsu.edu

[2]: Key Laboratory of Water Cycle and Related Land Surface Processes, Institute of Geographic Sciences and Natural
Resources Research, Chinese Academy of Sciences, Beijing, 100101, China; Email: liusq@igsnrr.ac.cn

*Correspondence to*: Renjie Zhou (renjie.zhou@shsu.edu)

**Abstract.** Arctic rivers represent important components of the Arctic and global hydrological and climate systems,
serving as dynamic conduits between terrestrial and marine environments in some rapidly changing regions. They
transport freshwater, sediments, nutrients, and carbon from vast watersheds to the Arctic Ocean and affect ocean
circulation patterns and regional climate dynamics. Despite their importance, modeling Arctic rivers remains
challenging because of sparse data networks, unique cryospheric dynamics, and complex responses to
hydrometeorological variables. In this study, a novel hybrid deep learning model is developed to address these
challenges and predict Arctic River discharge by incorporating Kolmogorov-Arnold Networks (KAN), Long Short-
Term Memory, and the attention mechanism with seasonal trigonometry encoding and physics-based constrains. It
integrates several novel components: 1) The KAN-based deep learning component learns and captures intricate
temporal patterns from nonlinear hydrometeorological data; 2) Explicit physical constrains designed for the
characteristics of permafrost-dominated watersheds govern snow accumulation and melt processes through the
architectural design and loss function; 3) The seasonal variations are accounted for using trigonometry functions to
represent cyclical patterns; 4) A residual compensation stricture allows the proposed model to revisit systematic
errors in initial predictions and helps capture complex nonlinear processes that are not fully represented. The
Kolyma River, which is significantly dominated by the permafrost, is adopted to test the performance of the newly
developed model. It obtains more robust and accurate predictive performance compared to baseline models. The role
of physical constraints, the residual compensated architecture, and the trigonometry encoding are assessed by
ablation analysis. The results indicate that these components positively contribute to improving the predictive
performance. This novel approach addresses the unique challenges of hydrological forecasting in cold, permafrost-
dominated regions and provides a robust framework for predicting Arctic River discharge under changing climate
conditions.

## 1. Introduction

Arctic rivers are integral to the Arctic's hydrological cycle and global climate systems and have undergone
significant changes in recent years (Rawlins and Karmalkar, 2024). They are essential for transporting vast amounts



of freshwater, sediments, and organic matters from terrestrial sources to the Arctic Ocean and sustaining the

biodiversity of the region and supporting unique ecosystems (Bring et al., 2016; Tank et al., 2023a; Vonk et al., 2025). The intricate connections between Arctic rivers and other cryospheric and atmospheric components make them highly sensitive to climate change (Feng et al., 2021). The response to climatic shifts, including changes in precipitation patterns, temperature regimes, snowmelt timing, and evapotranspiration rates in Arctic watersheds, has far-reaching implications for ecosystem stability and introduces significant uncertainties into future climate

projections (Peterson et al., 2002).

Predicting hydrodynamics of Arctic rivers remains challenging due to the region's unique environmental conditions, data scarcity, complex feedback mechanisms, and their nonlinear responses to temperature, rainfall, and evapotranspiration. For example, warming temperatures can accelerate permafrost thaw and alter hydrological cycles in Arctic regimes. Temperature thresholds play a crucial role, particularly around the 0°C mark, where phase

changes in precipitation and surface water create abrupt shifts in river dynamics (Prowse et al., 2011; Walvoord and Kurylyk, 2016). These temperature dependents transitions are further complicated by permafrost thawing, which destabilizes riverbanks, modifies groundwater flow paths, changes groundwater-surface water interactions, and increases sediment and nutrient loads, creating intricate feedback loops and complicates flow predictions (McClelland et al., 2004; Wang et al., 2021).

Over the last several decades, significant efforts have been directed towards forecasting the responses of river discharge to hydrometeorological conditions and understanding the underlying driving mechanisms (Gelfan et al., 2017; Jin et al., 2024a; Wang et al., 2021; Zhang et al., 2023; Zhou and Zhang, 2023a). These approaches can be broadly categorized into process-based models and empirical models. Process-based models simulate detailed physical and chemical processes within hydrological systems. For example, Gelfan et al. (2017) employed process-

based hydrological models, including the HYdrological Predictions for the Environment (HYPE) and ECOlogical Model for Applied Geophysics (ECOMAG), to simulate the hydrodynamics of the Lena and Mackenzie Rivers and assessed the impacts of climate change. Similarly, Krogh et al. (2017) developed a physics-based hydrological model that accounted for key hydrological processes for quantifying water losses at the tundra-taiga transition in a small Arctic basin. While these process-based approaches yield valuable insights into the underlying hydrological

processes and mechanisms, their successful implementation usually requires extensive parameterization and detailed characterization of environmental conditions, such as topography, spatially distributed hydrological parameters, and vegetation patterns. Such comprehensive data requirements pose significant challenges in Arctic regions, where remote locations, limited infrastructure, and harsh climatic conditions constrain field measurements and sustained monitoring campaigns (Gao et al., 2020). In contrast, empirical models, particularly data-driven approaches, focus

on establishing direct mappings between input and output variables without requiring comprehensive understanding of the underlying hydrological systems (Zhou and Zhang, 2022b).

Recently, data driven models have been increasingly developed and used to simulate hydrodynamics and characterize hydrological systems in Arctic regions. For instance, Zhang et al. (2023) simulated the streamflow changes of several major Arctic rivers with meteorological conditions using a Support Vector Regression model.

This machine learning model was then used to estimate responses of these rivers to the elevated temperature and





precipitation conditions. Singh et al. (2020) implemented several convolutional neural networks models (CNN), including UNet, SegNet, Deeplab and DenseNet, to estimate surface concentration of river ice. Their approach demonstrated improved estimation performance compared to existing methods by addressing the key challenge of noise and errors in the limited available training data. Sergeev et al. (2024) developed a hybrid model integrating
wavelet transform with long short-term memory (LSTM) networks for predicting Arctic methane concentration with greenhouse gases data monitored from the Belyy Island in Russia.

Despite these advances, significant challenges remain in modeling intricate river systems. Current deep learning approaches often struggle to capture complex and nonlinear relationships between meteorological variables and river discharge (Jin et al., 2024b; Zhou et al., 2024a). To improve the performance when dealing with nonlinear data such
as rainfall-runoff relationship, many technologies have been developed. For example, Basu et al. (2022) proposed a nonlinear autoregressive model with exogenous variables for flooding prediction in Ireland. Bakhshi Ostadkalayeh et al. (2023) used Kalman Filter (KF) to manage nonlinear systems and improve LSTM performance for forecasting streamflow. Zhou et al. (2024b) integrated the ensemble empirical model decomposition technology with temporal fusion transformers and developed a new hybrid deep learning model for discharge prediction, which outperformed
baseline models. Liu et al. (2024) proposed Kolmogorov-Arnold Networks (KAN) based on the theoretical foundation in the Kolmogorov-Arnold theorem. Unlike traditional neural networks that use fixed activation functions, the KAN model parameterized learnable activation functions on the connections between nodes, which significantly enhances the model's capacity to capture complex nonlinear relationships in data.

In addition, the scarcity of training data in Arctic regions limits the generalization of traditional deep learning
models, leading to less satisfying performance (Alzubaidi et al., 2023). Physics-informed neural networks (PINN) and physics-guided deep learning approaches offer a promising solution by incorporating physical constraints and domain knowledge into the learning process (Karniadakis et al., 2021). By embedding physical laws into the loss function, these hybrid approaches can improve prediction accuracy while ensuring physically consistent results (Zhong et al., 2024). A variety of physics-informed deep learning models have been developed and demonstrated
promising results in various hydrological applications. For example, Yang et al. (2020) proposed a hydrological model that integrated the physical process with a machine learning model for simulating daily streamflow. This hybrid model obtained accurate predictions for long-term daily streamflow with limited training data and demonstrated the effectiveness of this approach for reducing data requirements. Xie et al. (2021) integrated physical mechanisms into a deep learning model through both modified loss functions and synthetically generated training
samples for forecasting streamflow. Their model outperformed traditional models and highlighted the value of incorporating physical constraints into deep learning frameworks for hydrological modeling.

To address these challenges and improve predictive performance in permafrost-dominated Arctic rivers, a novel hybrid residual compensated deep learning model that integrated seasonal patterns, physics-based constraints, KAN, LSTM and attention is proposed for forecasting Arctic River discharge in this study. This newly proposed model
introduces several key innovations that serve specific purposes: (1) a KAN-based deep learning model coupled with LSTM and the attention mechanism, which enables sophisticated feature representation and temporal patterns recognition for nonlinear hydrometeorological data; (2) physical constrains that explicitly govern snow



accumulation and melt processes, which ensure physical consistency through the architectural design and loss function; (3) a residual compensation structure that combines a physics-informed main network with a specialized

residual network, which allows the model to capture physically governed patterns and local anomalies; and (4) a temporal pattern recognition system that incorporates cyclical encoding of seasonal features for seasonal variations. This integrated approach is specifically designed to address the challenges of hydrological forecasting in cold, permafrost-dominated regions, where snow accumulation and melt play a crucial role in seasonal discharge patterns. The innovative components are integrated to enhance its predictive accuracy, physical consistency, and ability to

handle complex seasonal dynamics and hydrological processes that characterize Arctic River systems.

## 2. Study area and data acquisition

To assess the performance, the newly developed model is tested on the Kolyma River located in the northeaster Siberia. The Kolyma River is one of the major Arctic rivers with a mean annual discharge of 136 km$^3$/year and the largest river system draining into the East Siberian Sea. The Kolyma watershed is Earth's largest watershed that is

120 100% underlain by continuous permafrost (Holmes et al., 2012). The extensive permafrost coverage makes the Kolyma watershed particularly sensitive to climate warming, leading to its unique hydrological behaviors (Spencer et al., 2015). With a drainage basin of approximately 647,000 km², the Kolyma River flows through diverse landscapes including the Kolyma Mountains, permafrost regions, and tundra ecosystems. The river's discharge regime is characterized by a distinctive seasonal pattern, with peak flows occurring during the spring snowmelt

period (May-June) and low flows during the winter months when the river is ice-covered (Bring et al., 2016). In this study, monthly temperature ($T$), precipitation ($P$) and potential evapotranspiration ($PET$) are used as input variables for forecasting discharge values of the Kolyma River. The Kolyma discharge records (1978-2020) at the Kolymsk gauge station (68.73°N, 158.72°E) are obtained from the ArcticGRO Discharge Dataset (Version 20231204). Note that the historical discharge data of the Kolyma River is not used as input variables in this study,

which allows the model to establish direct relationships between hydrometeorological drivers and river discharge without incorporating autoregressive components, thereby focusing specifically on how climatic factors influence discharge patterns in permafrost-dominated watersheds. Gridded monthly average 2-m temperature and potential evapotranspiration with a resolution of 0.5° are obtained from CRU TS v. 4.07 (Harris et al., 2020). Additionally, monthly precipitation data at a 0.5° resolution are obtained from the Global Precipitation Climatology Centre

(GPCC) dataset (Schneider et al., 2022). The complete dataset spans from January 1978 to December 2020, which is partitioned into training (80%) and testing (20%) datasets for model development and performance assessment.




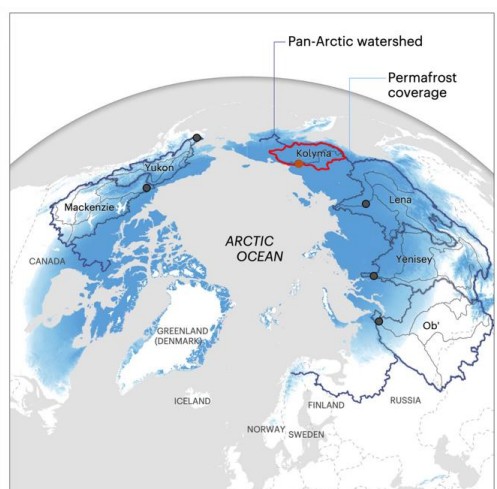

**Figure 1: The geographic location of the Kolyma River, modified from Tank et al. (2023b).**

## 3. Methodology

Hydrological forecasting in Arctic and permafrost-dominated regions presents unique challenges due to the strong influence of snow accumulation, permafrost thawing, and seasonal melt dynamics on river discharge. To address these issues, a novel hybrid residual-compensated physics-informed KAN-LSTM with attention model (RCPIKLA) that leverages strengths of multiple deep learning structures while embedding physical constraints related to snowmelt energy balance and seasonal variations directly into the training process for improved prediction accuracy and reliability.

As shown in Fig. 2, monthly precipitation, temperature and evapotranspiration data are preprocessed and standardized to ensure all features contribute appropriately to the training process. In regions dominated by permafrost, snow accumulation and melt typically exhibit strong seasonal periodicity (Andersson et al., 2021; Ernakovich et al., 2014). To include these cyclical patterns and facilitate smooth temporal transition, the month of the year is encoded using trigonometric transformations as $\text{Month}_{\sin} = \sin\left(\frac{2\pi m}{12}\right)$ and $\text{Month}_{\cos} = \cos\left(\frac{2\pi m}{12}\right)$, where $m$ refers to the month $m \in \{1, 2, ..., 12\}$. The trigonometric features are concatenated with other input variables, including temperature, precipitation and evapotranspiration, and fed into the residual-compensated physics-informed KAN-LSTM model with attention. The newly proposed model leverages the KAN component as a feature transformation layer to extract and learn complex nonlinear patterns from hydrological and meteorological datasets. The LSTM component captures short- and long-term dependencies and effectively simulates sequential patterns and discharge variability. To further refine temporal learning, the attention mechanism is introduced and integrated, which allows the proposed model to selectively emphasize historically significant time steps, particularly those driving major and seasonal hydrological transitions. An important innovation is the residual compensation structure, which explicitly addresses the challenges of predicting extreme discharge events. By learning systematic



error patterns, the residual structure can adjust simulations based on residual predictions and improve performance during high-variability sceneries. Unlike conventional data-driven models that ignore fundamental physical

constraints, the newly developed model incorporates physics-informed loss functions, ensuring that snow accumulation and melt timing adhere to thermodynamic energy balance principles. Additionally, the model employs seasonality-aware encoding using trigonometric transformations to recognize the cyclic nature of hydrological processes. This architecture is designed to provide an accurate and robust framework for forecasting river discharge in Arctic and permafrost-dominated environments.

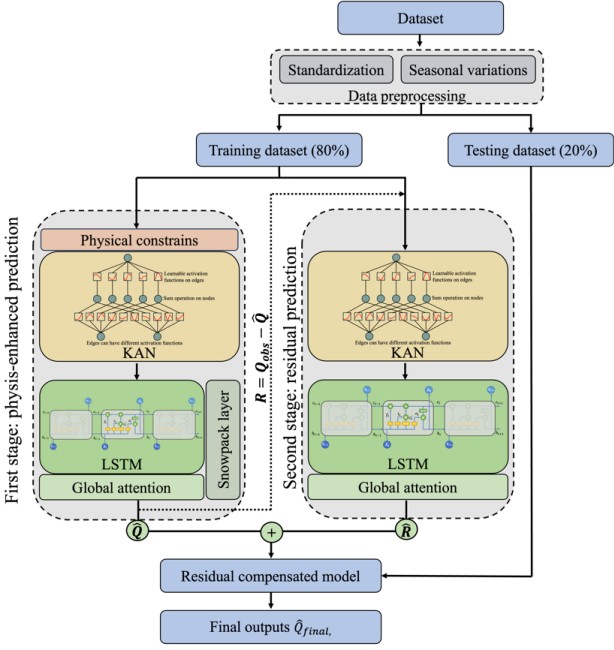

**Figure 2: The architecture of the residual compensated physics-informed KAN-LSTM model with attention.**

### 3.1. Kolmogorov-Arnold Networks

In the Kolmogorov-Arnold representation theorem, it states that any continuous multivariate function can be represented as a superposition of continuous functions of a single variable (Kůrková, 1992). Based on this theoretical foundation and the mechanism of decomposing the multivariate function into various univariate functions, the Kolmogorov-Arnold Networks model (KAN) was developed by replacing all weight parameters with

univariate functions parameterized as splines, rather than using Multi-Layer Perceptrons (MLPs) in traditional neural networks (Liu et al., 2024). This structure allows the KAN model to dynamically adapt its processing to various aspects of the data and emphasize finer details by modulating the granularity of these splines (Granata et al., 2024). With learnable activation functions and structured transformations, it can effectively extract nonlinear



relationships and capture intricate patterns, making it well-suited for modeling complex hydrological systems like
Arctic River discharge.

In this newly developed hybrid model, the KAN module is used as an advanced feature transformation block and a nonlinear feature extractor that processes raw hydrological and meteorological inputs before the sequential modeling stage. The architecture of the KAN module is composed of several parts: 1) input expansion: the raw input features including precipitation, temperature and evapotranspiration are first projected into a higher dimensional space by a
fully connected layer that increases the representational capacity. The dimension expansion of the input features allows the model to isolate some nonlinear interactions between variables, such as temperature-driven snowmelt thresholds or precipitation-phase transitions; 2) Nonlinear activation: a Gaussian Error Linear Unit (GELU) activation is then applied to the expanded features. The GELU function introduces smooth nonlinearity and enables the network to capture intricate patterns in the input data, which approximates the role of univariate functions in the
Kolmogorov-Arnold theorem while avoids the computational overhead of spline optimization; 3) Dimensionality reduction: a second linear layer then compresses the activated features down to a lower-dimensional space which is then fused with physics-based constrains, such as snowpack dynamics and fed into the LSTM-Attention network for temporal integration. It aims at effectively distilling the information into a compact, yet expressive representation that is more amenable for subsequent processing. The KAN transformation and processing steps can be expressed as
the following equations accordingly:

$$H_1 = W_1 X + b_1, \tag{1}$$

$$H_2 = GELU(H_1), \tag{2}$$

$$KAN(X) = W_2 H_2 + b_2, \tag{3}$$

where $X$ is the input features; $W_1$ and $W_2$ refer to the expansion and compression weight matrices; $b_1$ and $b_2$ are the
corresponding bias vectors; GLUE is the Gaussian Error Linear Unit activation function.

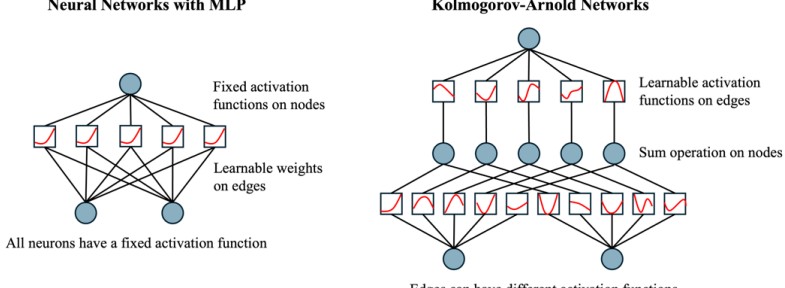

**Figure 3: The structure of Kolmogorov-Arnold Networks (KAN) compared to MLP.**

### 3.2. Long Short-Term Memory

Following the Kolmogorov-Arnold transformation, the processed input features will enter the Long Short-Term Memory (LSTM) module. LSTM is a modified variant of recurrent neural networks (RNNs), specifically designed to address the vanishing gradient problem while learning long-term dependencies in sequential data (Hochreiter and
Schmidhuber, 1997). By incorporating the gating mechanism and a hidden state, LSTM can efficiently regulate



information flow through the network and selectively remember or forget information in long sequences. Because of its ability to capture temporal dependencies inherent in river systems, the LSTM model has been widely used in a variety of hydrological models (Gao et al., 2020; Zhou and Zhang, 2023b). It aims at learning and identifying important historical patterns in meteorological variables (such as temperature and precipitation) that influence

current river discharge, while simultaneously recognizing the varying time lags between these inputs and their hydrological responses. This capability makes LSTMs especially suitable for modeling Arctic River systems, where discharge patterns are influenced by both immediate meteorological conditions and longer-term processes such as snowmelt and permafrost dynamics (Kratzert et al., 2018).

The memory cell of LSTM is primarily composed of three gates: the input gate ($i_t$), forget gate ($f_t$), and output gate

($o_t$). The input gate determines which new information should be stored in the cell state, while the forget gate decides what information should be discarded from the previous cell state. The output gate controls how much of the cell state should be exposed to the next layer. This gating mechanism allows LSTMs to maintain and update relevant information over long sequences while filtering out irrelevant details (Hochreiter and Schmidhuber, 1997). At any time step $t$, the hidden state ($h_t$) and the cell state ($c_t$) are calculated based on the previous hidden state ($h_{t-1}$) and cell

state ($c_{t-1}$) with three logic gates as follows:

$$f_t = \sigma\left(W_f X_t + U_f h_{t-1} + b_f\right), \tag{4}$$

$$i_t = \sigma(W_i X_t + U_i h_{t-1} + b_i), \tag{5}$$

$$c'_t = tanh(W_c X_t + U_c h_{t-1} + b_c), \tag{6}$$

$$c_t = f_t \otimes c(t-1) + i_t \otimes c'_t, \tag{7}$$

$$o_t = \sigma(W_o X_t + U_o h_{t-1} + b_o), \tag{8}$$

$$h_t = o_t \otimes tanh(c_t), \tag{9}$$

Where $c_t$, $c'_t$, and $h_t$ are the cell state, candidate cell state, and hidden state at time step $t$, respectively; $X_t$ refers to the input variables processed by the KAN module; $W$, $U$ and $b$ are weight matrices and bias vectors whereas subscripts $f$, $i$, $c$, and $o$ denote the forget gate, input gate, candidate cell, and output gate; $\sigma$ and $tanh$ are the sigmoid

and hyperbolic tangent activation functions; $\otimes$ is the element-wise operation.

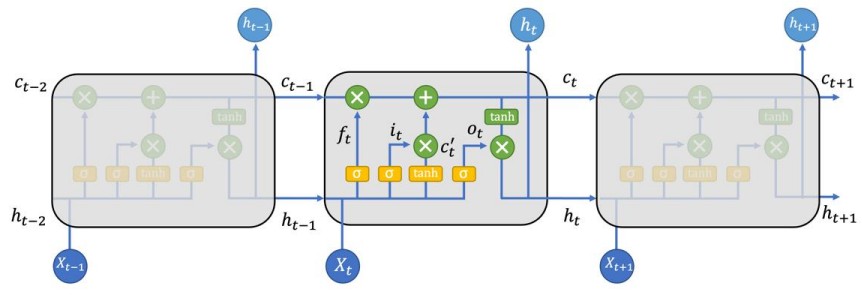

**Figure 4: The architecture of the LSTM model.**

**3.3. Attention**



A global attention mechanism is incorporated into the LSTM component of the newly proposed model to assign different importance weights to past time steps when making predictions, which enables the model to dynamically weight and aggregate information across temporal sequences. As the influence of historical conditions on current

discharge exhibits complex temporal dependencies in hydrological modeling, the attention mechanism can help capture both short-term fluctuations and long-range interactions in input variables. The attention score for each time step can be computed as (Vaswani et al., 2017):

$$e_t = v^T \tanh(W_a h_t + b_a), \tag{10}$$

$$\alpha_t = \frac{\exp(e_t)}{\sum_j \exp(e_j)}, \tag{11}$$

$$C = \sum_t \alpha_t h_t, \tag{12}$$

$$\hat{Q} = W_c C + b_c, \tag{13}$$

where $W$ and $b$ denote weight and bias parameters; $e_t$ refers to the attention score at time step $t$; $h_t$ is the hidden state from the LSTM component at time step $t$; $v$ is a learnable vector which determines the importance of each hidden state; $\alpha_t$ is the attention weight; $C$ is the context vector that represents a weighted sum of all hidden states; $\hat{Q}$

refers the discharge prediction using the context vector calculated from the context vector.

### 3.4. Physics-informed mechanisms

Physics-informed neural networks improve hydrological modeling by combining established physical information with deep learning architectures, which creates a synergistic approach that leverages the strengths of both methodologies. In this study, a hybrid physics-informed approach is implemented through two complementary

mechanisms: 1) a dedicated snowpack layer directly integrated into the model architecture, and 2) a physics-constrained loss function. The snowpack layer explicitly simulates snow accumulation and melting processes based on temperature and precipitation. It tracks precipitation falling as snow when temperatures drop below freezing ($T <$ 0°C, where $T$ represents temperature) and computes snowmelt using a temperature-dependent rate function (Hock, 2003):

$$M_r = f_m \cdot \max(T, 0), \tag{14}$$

where $M_r$ is the melting rate, and $f_m$ is the melting factor coefficient. The melting factor of 0.5 mm/°C/day is adopted in this study based on empirical studies of Arctic snowpack dynamics (Hock, 2003). The snowpack mass balance is estimated as follows (DeWalle and Rango, 2008):

$$S_t = S_{t-1} + P_t^{snow} - M_t, \tag{15}$$

where $S_t$ and $S_{t-1}$ denote the snowpack water equivalent at time $t$ and $t$-1; $M_t$ is the actual snowmelt, which is calculated as $M_t = \min(S_{t-1}, M_r)$; $P_t^{snow}$ refers to the snowfall fraction of precipitation, which is determined by the following equation (Harpold et al., 2017):

$$P_t^{snow} = \begin{cases} P_t, & \text{if } T < 0°C \\ 0, & \text{otherwise} \end{cases}, \tag{16}$$

where $P_t$ is the precipitation rate. A key architectural innovation is that the calculated snowmelt amount is directly

added to the data-driven neural network output before the final activation function of the first stage as shown in Fig. 2, creating a hybrid prediction that leverages both physical understanding and learned patterns:



$$\hat{Q} = \text{ReLU}(Q_{LSTM} + M_t), \tag{17}$$

Where $\hat{Q}_i$ and $Q_{LSTM}$ are the predicted discharge from the first stage and the output from the LSTM with attention component, respectively; ReLU refers to rectified linear unit activation function.

In addition to the snowpack layer, a physics-constrained loss function is implemented for enforcing physical consistency through the term:

$$\mathcal{L}_{phys} = \frac{1}{n}\sum_i max(M_t - \hat{Q}_i, 0), \tag{18}$$

where $n$ is the number of samples, and $\mathcal{L}_{phys}$ refers to the physics-constrained loss function term. This term penalizes physically inconsistent predictions where the modeled discharge is less than the calculated snowmelt contribution. This dual physics-guided approach is particularly valuable for Arctic rivers where seasonal snow accumulation and permafrost melt dominate the hydrological regime. In these regions, river discharge often exhibits complex, threshold-dependent behaviors and memory effects related to temperature-controlled phase changes in water, processes that purely statistical models often struggle to capture accurately without explicit physical constraints. By incorporating both a direct snowmelt contribution mechanism and physics-consistency loss penalties, the proposed model maintains physical realism even when data limitations exist.

### 3.5. Residual compensated mechanism

While the physics-informed deep learning model may improve prediction accuracy by embedding domain knowledge, they may still fail to capture certain discrepancies between observed and predicted discharge values caused by sources, such model simplifications, missing hydrological processes, noise in the input data, and extreme events. To address this limitation, a residual compensated mechanism is incorporated. As shown in Fig. 2, the residual compensated framework in the newly proposed model operates in a two-stage process. First, we train a physics-informed KAN-LSTM model that incorporates snowpack dynamics and constraints through the combined loss function ($\mathcal{L}_{combined}$):

$$\mathcal{L}_{combined} = \alpha\mathcal{L}_{MSE}(\hat{Q}, Q_{obs}) + \beta\mathcal{L}_{phys}, \tag{19}$$

where $\mathcal{L}_{MSE}$ refers to the mean squared error between the prediction $\hat{Q}$ and the observation $Q_{obs}$; $\alpha$ and $\beta$ are weighting coefficients which can be obtained by trial-and-error. In the second stage, the residuals ($R_i$) between observations and physics-based predictions are computed: $R_i = Q_{obs,i} - \hat{Q}_i$. These residuals represent the information discrepancies that the physics-informed KAN-LSTM model fails to capture. A separate residual model ($M_{res}$) which has a KAN-LSTM architecture without physics-informed components is trained to specifically learn the discrepancies: $\hat{R}_i = M_{res}(X_i)$. The final discharge prediction ($\hat{Q}_{final,i}$) is obtained by combining results from the first and second stage:

$$\hat{Q}_{final,i} = \hat{Q}_i + \hat{R}_i. \tag{20}$$

This residual compensated approach has several advantages: on one hand, it preserves the physical consistency by incorporating the physics-informed component during the first stage. On the other hand, the residual prediction in the second stage can focus exclusively on missed patterns and systematic anomalies, creating a specialized representation for complex processes. As a result, it enables end-to-end training where each component focuses on




complementary aspects of the hydrological system: the physics-informed deep learning model captures the first-order processes driven by hydrometeorological variables, while the residual model captures secondary influences and complex feedback mechanisms. It is especially benefit for Arctic River systems, where seasonal transitions and complex cryospheric processes may not be fully captured by simplified physics representations.

### 3.6. Evaluation metrics

To assess the performance of the proposed model in the Kolyma River, two popular evaluation metrics are adopted in this study: Nash-Sutcliffe Efficiency (NSE) and Root Mean Square Error (RMSE) (An et al., 2020; Zhou and Zhang, 2022a). NSE is a dimensionless metric widely used in hydrological modeling that measures how well the model predictions match the observed data compared to using the mean of the observations as a predictor. An NSE value of 1 indicates a perfect fit, while values approaching zero or negative suggest that the model performs no better than using the mean value of the observed data. The NSE value can be calculated as:

$$NSE = 1 - \frac{\sum_{i=1}^{n}(Q_{obs,i} - \hat{Q}_{final,i})^2}{\sum_{i=1}^{n}(Q_{obs,i} - \bar{Q})^2},\tag{21}$$

where $Q_{obs,i}$ and $\bar{Q}$ are the observed discharge value at time step $t$ and the average discharge, respectively. In hydrological modeling, NSE values above 0.75 indicate very good model performance (D. N. Moriasi et al., 2007). RMSE is an absolute error metric that quantifies the average magnitude of prediction errors in the original units of discharge being predicted. RMSE gives higher weight to large errors due to its squared terms, which makes it particularly useful for evaluating models where large errors are especially undesirable, such as in flood prediction. Lower RMSE values indicate better model performance, with RMSE = 0 representing a perfect fit. It is defined as:

$$RMSE = \sqrt{\frac{1}{n}\sum_{i=1}^{n}(Q_{obs,i} - \hat{Q}_{final,i})^2}.\tag{22}$$

These two metrics complement each other in our evaluation framework. While NSE provides a normalized measure that facilitates comparison across different time periods, RMSE provides an intuitive measure of error magnitude in the original units. Together, they provide a comprehensive assessment of the model's ability to capture both the temporal dynamics through NSE and the absolute accuracy through RMSE of river discharge predictions in the Kolyma River system.

## 4. Results

### 4.1. Performance comparison among various baseline models with various time steps

The newly proposed model and baseline models are trained in the training dataset of the Kolyma River, and then the fine-tuned models are applied to the unseen testing dataset for the assessment of the predictive performance. The model performance across different time steps (1-12 months) reveals variations in predictive capabilities among the models tested. To ensure stable results, each model is run 10 times at each time step, and the evaluation metrics are averaged. Presented in Fig. 5, it shows the comparison of NSE values (Fig. 5 Left) and RMSE values (Fig. 5 Right) for the Kolyma River discharge predictions using several different model architectures, which include the simple RNN, LSTM, and GRU models, which are popular temporal baseline models widely used in many hydrological




studies, and the newly proposed Residual Compensated Physics-Informed KAN-LSTM with Attention (RCPIKLA).
The NSE values demonstrate that the newly proposed RCPIKLA model consistently outperforms all baseline models
across all time steps, achieving the highest NSE values ranging from 0.78 to 0.86. This superior performance is
particularly obvious at the time step of 9 months, where RCPIKLA reaches peak NSE values of approximately 0.86.
The traditional deep learning models, including the simple RNN, GRU, and LSTM models, show similar

performance patterns with NSE values ranging between 0.65 and 0.76. These models exhibit a noticeable decline in
performance at medium-range time steps (4-8 months), with their lowest NSE values observed around months 5-6,
which suggests limitations in capturing seasonal transitions in Arctic River systems. The RMSE analysis
corroborates these findings, with RCPIKLA achieving the lowest error values (6.5 mm -8.5 mm) across all time
steps. Again, the RCPIKLA model demonstrates substantially lower prediction errors compared to other baseline

approaches, which exhibit RMSE values ranging from 9.5 mm to 11.5 mm. The higher RMSE values for Simple
RNN, GRU, and LSTM at medium-range time steps further highlight their difficulties in accurately predicting
discharge during critical seasonal transition periods.
To further evaluate the robustness and generalization ability of each model, we conduct the box plots of 10
independent runs for each architecture and compute the distributions of NSE and RMSE across all time steps. These

box plots, as shown in Figure 7, provide insight into the statistical variability and stability of model performance.
The RCPIKLA model demonstrates the best overall performance with the highest median NSE and lowest median
RMSE, along with the narrowest interquartile range. This indicates not only high accuracy but also low variability
across runs, suggesting a stable learning and prediction process. Moreover, outliers are less frequent and less
extreme for RCPIKLA, which indicates a consistently reliable model output. LSTM and Simple RNN exhibit greater

variance in both NSE and RMSE distributions, with wider interquartile ranges and more outliers. This means higher
sensitivity to random initialization and potential overfitting or underfitting in different runs. GRU shows moderately
better consistency than LSTM and Simple RNN but still falls short of the stability achieved by RCPIKLA. The
newly proposed RCPIKLA model consistently outperforms all other models across different time steps and obtains
robust performance. These results demonstrate that incorporating physical constraints with the KAN-LSTM model

and complementing them with residual learning significantly improve predictive performance for capturing complex
patterns in Arctic River discharge.

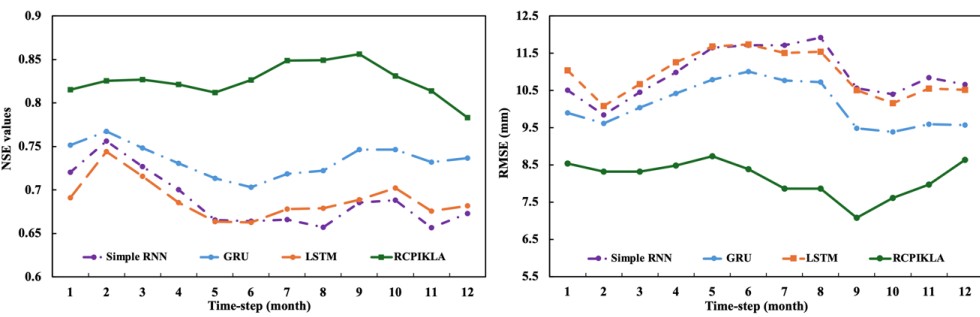





**Figure 5: NSE (left) and RMSE (right) values of multiple models over various time steps. The models include the residual-compensated physics-informed KAN-LSTM model with attention (RCPIKLA), simple RNN, LSTM, and GRU.**

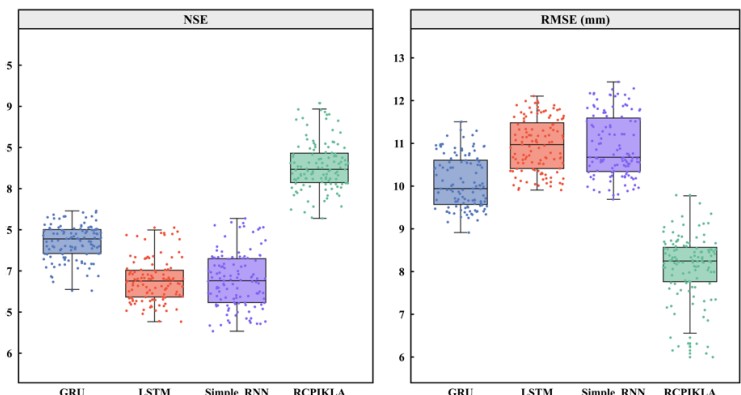

**Figure 6: The box plot of NSE (left) and RMSE (right) values of multiple models of various time steps (1-12 months) for 10 runs.**

### 4.2. Performance comparison among various deep learning models at different value ranges

As shown in Fig. 5, the optimal performance of the proposed RCPIKLA model is obtained when the time step is 9 months. In addition to temporal comparisons, the predictive performance across different discharge value ranges is further assessed to understand how well each model captures the full spectrum of hydrological variability. The predicted and observed values of the proposed model and baselines when the time step is 9 months are presented in Fig. 7. The red dash line angled at 45 degrees represents the line of perfect agreement between observed and predicted values. The performance metrics reveal substantial differences in model accuracy. The RCPIKLA model demonstrates more robust performance compared to others across all value ranges with the highest NSE coefficient of 0.856 and the lowest RMSE of 7.077 mm. This indicates that the proposed hybrid approach, which integrates physics-informed constraints with residual compensation, captures the nonlinear and non-stationary characteristics of the Kolyma River discharge more effectively than other architectures. The GRU model achieves an intermediate performance level (NSE = 0.750, RMSE = 9.418 mm), which overperforms other recurrent neural networks but falling short of KNN based models. Both LSTM and Simple RNN exhibit similar and relatively poorer performance metrics, which demonstrates their limitations in capturing the complex hydrological dynamics of Arctic River systems when used without additional enhancements.

It is worthwhile to note that all models perform reasonably well for low to moderate discharge values (0-30 mm), but significant differences emerge at higher discharge events (>80 mm), which is crucial for flood forecasting. Although the proposed RCPIKLA model maintains better prediction accuracy for these high discharge events, there is room for improvement, which may be attributed to the limited number of high discharge events in the training





dataset. This systematic underestimation of peak flows represents a common challenge in hydrological modeling of Arctic rivers, where extreme discharge events are relatively rare but carry significant implications for water resource management and hazard mitigation. Future work could address this limitation through specialized sampling

techniques or physics-informed constraints specifically designed to better capture high-magnitude discharge events.

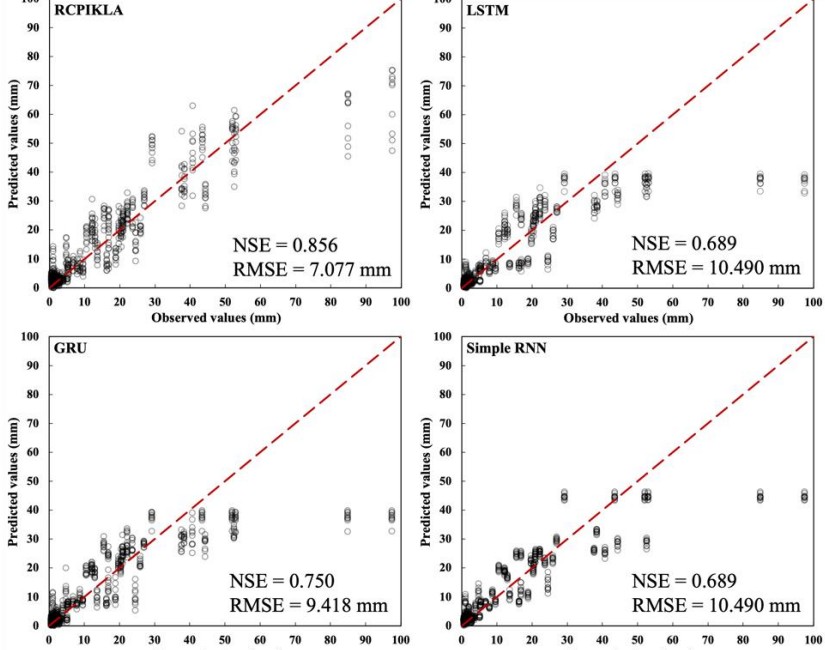

**Figure 7: The predicted and observed values of multiple models when the time step is 9 months, including RCPIKLA, LSTM, GRU and Simple RNN models.**

### 4.3. On the role of the physics informed constrains and residual structure

To isolate and evaluate the contribution of the physics-informed constraints and the residual learning structure, ablation experiments, which have been adopted by many other studies (Zhi et al., 2023; Zhou, 2025), are conducted using three model variants in Fig. 8, including the complete RCPIKLA model (incorporating both physics-informed

constraints and residual compensation), RCKLA-no physics-informed (retaining the residual structure but without physics constraints), and PIKLA-no residual (including physics-informed constraints but without residual compensation). The boxplot summarizes the NSE and RMSE values of three models for time steps ranging from 1 to 12 months in 10 independent runs. It reveals that the complete RCPIKLA model achieves the highest median NSE performance (approximately 0.83) with an interquartile range spanning from 0.81 to 0.84. The RCKLA model

without physics-informed constraints shows a slightly lower median NSE (approximately 0.81) with greater variability in interquartile range from 0.79 to 0.83. The PIKLA model without residual compensation demonstrates the lowest median NSE performance (approximately 0.79) with an interquartile range from 0.78 to 0.81. The distribution of RMSE value is consistent with NSE values. These comparative results highlight two important



aspects of the model architecture:1) The physics-informed constraints contribute to overall model robustness and performance stability. By incorporating physical principles of snowpack accumulation and melt processes through the specialized SnowpackLayer, the model better captures the underlying hydrological dynamics of the Arctic River system. The physics-informed loss function, which mathematically enforces the relationship between melted snow and discharge, helps maintain physical consistency in the predictions. 2) The residual compensation mechanism addresses model inadequacies by learning the systematic errors in the physics-based predictions. This is particularly valuable for handling complex nonlinear processes that are not fully captured by the simplified physical representations. The performance difference between PIKLA and RCPIKLA demonstrates that the residual structure successfully compensates for approximation errors in the physics-informed component. The synergistic integration of both components yields a new structure that balances data-driven flexibility with physical consistency. This hybrid approach is particularly advantageous in data-limited environments like Arctic Rivers, where the physics-informed constraints and the residual compensation help overcome model simplifications and data uncertainty.

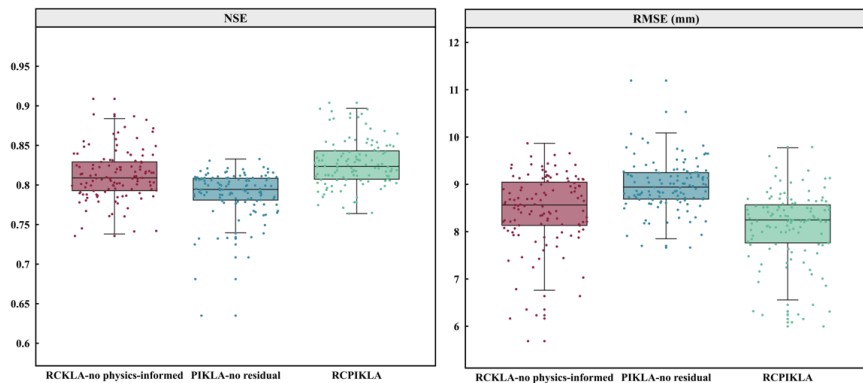

**Figure 8: The role of the residual structure and physics-informed constrains.**

### 4.4. the role of seasonal variations and trigonometric encoding

Seasonality plays a significant role in Arctic hydrological systems (Häkkinen and Mellor, 1992), where discharge patterns are strongly influenced by annual cycles of temperature, snow accumulation, and melt. Accurately capturing such periodic behaviors is essential for robust long-term forecasting models. To address this, a trigonometric encoding (TE) of seasonal features is incorporated as input variables using sine and cosine transformations of the calendar month. Specifically, the timestamp is mapped to two features using the following equations:

$$\text{Month}_{\sin} = \sin\left(2\pi\frac{M}{12}\right); \text{Month}_{\cos} = \cos\left(2\pi\frac{M}{12}\right), \tag{23}$$

where $M$ refers to the calendar month. These encodings aim at capturing cyclical temporal patterns without introducing artificial discontinuities between December and January. They are added to the input feature set of all models, which allows the proposed model to better associate temporal patterns with hydrometeorological signals.



As shown in Fig. 9, the box plot compares NSE and RMSE distributions for multiple model variants with and without trigonometric encoding (TE) of monthly seasonality. The results demonstrate that trigonometric encoding substantially improves performance across all model architectures. The proposed RCPIKLA model maintains the highest median NSE (approximately 0.83) with trigonometric encoding, while the removal of TE (denoted by "-no TE") leads to degraded performance (median NSE around 0.80) and wider value ranges. This pattern is consistent across all architectures, with GRU, LSTM, and Simple RNN models all exhibiting substantial performance degradation when seasonal encoding is removed. The widths of the box plots, representing interquartile ranges, also decrease substantially with TE, indicating greater consistency and reduced variability across model runs. Similar improvements are observed in GRU, LSTM, and Simple RNN models. In particular, the LSTM and Simple RNN models without trigonometric encoding show greater instability, with some runs achieving NSE values below 0.5, which shows severely compromised predictive capability. Regarding RMSE, the incorporation of TE effectively reduces median errors and decreases variability, particularly for RCPIKLA, where RMSE values exhibit the narrowest range. Outliers observed in models without trigonometric encoding suggest that omitting seasonal encodings can lead to occasional severe prediction errors, likely caused by the model's inability to account effectively for seasonal patterns.

Overall, the strong performance degradation observed when removing trigonometric encoding indicates the strong seasonality of Arctic River discharge. This seasonality can be characterized by processes including winter low flow due to frozen conditions, spring peak flow during snowmelt, and moderate summer flows influenced by rainfall and evapotranspiration. Without explicit encoding of this cyclical pattern, models struggle to establish accurate temporal context for the meteorological inputs, resulting in compromised predictive accuracy.

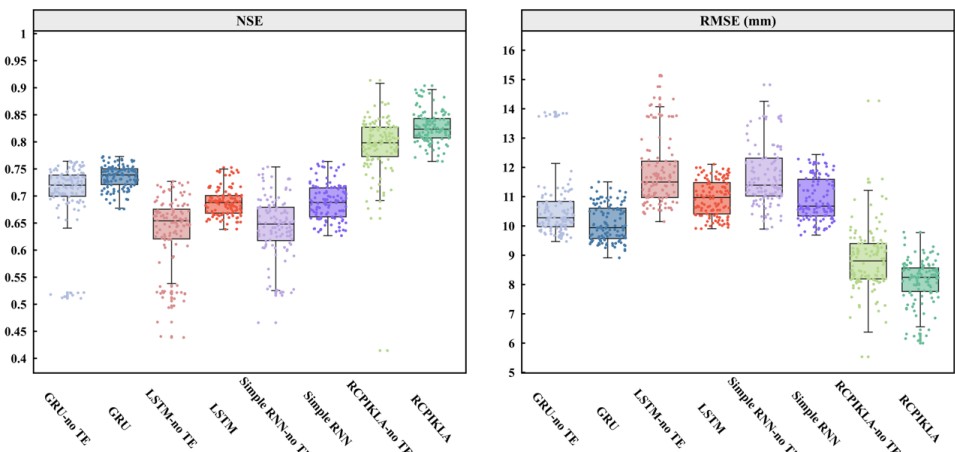

**Figure 9. The comparison of models with and without trigonometric encoding for seasonal variations as inputs.**

**5. Conclusion**



Arctic river systems play a critical role in global climate regulation, carbon cycling, and regional ecosystems, yet they remain challenging in hydrological modeling due to sparse data networks, the complex dynamics of permafrost-dominated landscapes, and nonlinear characteristics. In this study, a novel hybrid residual compensated deep learning model is proposed and designed specifically for Arctic River discharge forecasting. By integrating Kolmogorov-Arnold Networks (KAN), long short-term memory (LSTM) and the attention mechanism with seasonal encoding and physics-based constraints, the newly proposed approach aims at addressing the unique challenges of hydrological forecasting in permafrost-dominated regions, making it particularly valuable for modeling complex seasonal dynamics in Arctic River systems where snow accumulation and melt dominate discharge patterns. The KAN structure leverages the strengths of learnable activation functions and structured transformations to effectively extract nonlinear and intricate patterns from the input data. The newly proposed model is applied to the Kolyma River and compared with several baseline models, including LSTM, GRU, and simple RNN model for assessing the role and contribution of various components. Future applications could extend to other Arctic watersheds, snow-dominated river systems in mid-latitudes, and potentially other environmental domains. The results are summarized as follows:

1). The predictive performance of the newly proposed model and baseline models are plotted and evaluated across a range of time steps, from 1 to 12 months. As illustrated in Fig. 5, the newly proposed model consistently overperforms other baselines at all time steps and produces robust predictive performance. It obtains the highest NSE values ranging from 0.78 to 0.86 and the lowest RMSE values between 6.5 mm to 8.5 mm. The model performs the best at a time step of 9 months, suggesting that the permafrost covered Arctic River discharge exhibits a relatively long memory or delayed response to preceding hydrometeorological conditions.

2). The predictive performance across different discharge value ranges is further assessed to understand how well each model captures the full spectrum of hydrological variability. All models perform reasonably well for low to moderate discharge values (0-30 mm), but more obvious differences emerge at moderate and high discharge events. Although the proposed RCPIKLA model maintains improved prediction accuracy, challenges remain in accurately predicting extreme high discharge events, with all models showing a tendency to underestimate peak flows. This limitation may be partially attributed to the relatively sparse representation of high discharge events in the dataset, which constrains the model's ability to generalize under extreme hydrological scenarios.

3). Both physics-informed constraints and residual compensation contribute distinctly to model performance. The physics-informed component, which incorporates snowpack accumulation and melt processes, provides the proposed model with basic domain knowledge that helps overcome data limitations in the permafrost-dominated Kolyma River basin. The residual compensation mechanism examines systematic errors in the physics-based predictions and helps capture complex nonlinear processes that are not fully represented.

4) By transforming month values into sine and cosine components that preserve the cyclical nature of seasonal patterns, the incorporation of trigonometric seasonal encoding can improve the predictive performance. This approach enhances prediction accuracy across all architectures, with improvements of 4-6% in performance metrics, highlighting the importance of representing the pronounced seasonal dynamics of Arctic rivers characterized by



frozen winter conditions, spring snowmelt peaks, and moderate summer flows. The trigonometric seasonal is particularly effective when combined with the RCPIKLA architecture.

**Declaration of competing interest**

The author declares that he has no known competing financial interests or personal relationships that could have appeared to influence the work reported in this paper.

**Code/Data availability**

Data and code will be made available on request.

**Author contribution**

RZ: Writing – original draft, Visualization, Validation, Resources, Methodology, Investigation, Formal analysis, Data curation, Conceptualization. SL: Writing – review & editing, Resources, Data curation, Investigation.

**Acknowledgement**

This research was supported by the U.S. National Science Foundation (Award# 2407963).

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
