# Peer review of "A hybrid Kolmogorov-Arnold networks-based model with attention for predicting Arctic River streamflow"

_EGUsphere, 2025_

## Referee Comment (RC2)

**Manuscript:** A hybrid Kolmogorov-Arnold networks-based model with residual compensation and physics-informed constraints for Arctic River discharge prediction

**Journal:** Hydrology and Earth System Sciences (HESS)

**Comments:**

1. One of the primary advantages of Kolmogorov-Arnold Networks is their enhanced interpretability compared to traditional Multi-Layer Perceptrons (MLPs). KAN is usually used to improve the interpretability of the relations between inputs and output, but there is no mention of that.
The manuscript fails to leverage or discuss this fundamental strength of KAN architecture. Specifically, there is no:

• Visualization of the learned univariate functions

• Symbolic regression analysis

• Interpretation of what relationships the KAN component discovered between hydrometeorological inputs and Arctic discharge

• Physical insights into the processes governing snowmelt-driven streamflow in permafrost regions

Include a dedicated subsection on KAN interpretability analysis containing:

- Visualization of learned activation functions for key input-output relationships
- Symbolic approximations of these functions where feasible (using symbolic regression tools available in KAN libraries)
- Physical interpretation of discovered patterns in the context of Arctic hydrology
- Comparison with known physical relationships in snowmelt hydrology from the literature

2. what are the hyperparameters (epochs, batch size, learning rate) and details of the architecture of the RNN, GRU and other neural nets used for comparison.
The manuscript lacks essential details for all baseline models (RNN, GRU, LSTM):

- No specification of hyperparameters (epochs, batch size, learning rate)
- No architectural details (number of layers, hidden units, activation functions)
- No information about initialization methods
- No training procedure details (optimizer type, learning rate schedules, dropout rates)
- No stopping criteria or early stopping procedures
- No hardware specifications or training times

3. Recent papers suggest that KAN based architectures outperform classical ANN based architectures. There should have been a comparison with KAN based LSTM, GRU and other neural nets. The manuscript only compares RCPIKLA (which uses KAN) against traditional ANN-based models (RNN, GRU, LSTM), not against KAN-enhanced versions of these baseline architectures.

The comparison with no physics informed constraints and no residual has been compared. However, the current experimental design still creates an attribution problem. Observed performance improvements could stem from:

- The KAN component specifically
- The attention mechanism
- The physics-informed constraints
- The residual compensation structure
- Seasonal trigonometric encoding
- Some synergistic combination of these components

Without proper ablation comparing LSTM-attention/KAN-LSTM/KAN-GRU versus RCPIKLA, the specific contribution of KAN remains unclear.

4. The manuscript describes a physics-informed constraint that imposes an upper limit on predicted snowmelt contribution but does not explain the asymmetric treatment of constraint violations.

The asymmetric design requires clear physical justification:

- Upper bound rationale: Snowmelt contribution physically cannot exceed available snow water equivalent - this is a hard constraint based on mass conservation
- Lower bound question: Are underpredictions physically plausible? Could incomplete melting, refreezing, or sublimation make them valid? Or do they indicate model failure to capture melt processes?
- Bias implications: Does the asymmetric penalty introduce systematic bias toward underprediction?

5. Physics-informed neural networks fundamentally rely on balancing multiple loss terms through weighting parameters. The manuscript mentions $\alpha$ and $\beta$ as weights for MSE loss and physics loss but does not report their values.

The manuscript must provide:

- Final $\alpha$ and $\beta$ values used for all reported results
- Scenarios of hit and trials
- Search space explored

6. The manuscript lacks visualization of epoch-wise loss decomposition, which is important for assessment of convergence of all models. Without this analysis, it is impossible to assess whether the physics constraint meaningfully guides training or becomes negligible compared to the data-driven MSE loss.

Visualizing separate loss components reveals:

- Whether physics loss actually contributes to training or is overwhelmed by MSE loss

- Training stability and convergence behavior

- Potential issues: loss spikes, plateaus, phase transitions

7. Figure 6 (left): "y axis seems to be cut, the numbers are partly missing" - this affects readability and interpretation. Also, please check for spelling and grammatical errors throughout manuscript. Like a few spelling mistakes have been observed in abstract.

8. The physics-informed mechanism involves snow storage ($S_t$) and melt ($M_t$) terms that evolve over time. However, the manuscript does not specify:

- Initial values for $S_0$ and $M_0$ at the start of the simulation period
- How these initial conditions were integrated into the model?

9. It is mentioned conducting 10 independent runs but provides unclear or incomplete reporting of variability in results. Fig8 represents the rmse and nse RCPIKLA variants with all predictions, what is the average RMSE over 10 runs, how much variation is observed over independent runs?
Additionally:

• Figure 8 shows results (RMSE and NSE for RCPIKLA variants) but it's unclear whether these represent single runs, mean values, or distributions

• No explicit reporting of mean ± standard deviation for performance metrics

• No statistical significance testing comparing model variants

10. Figure 5 currently shows model predictions at 12 time intervals (representing different aggregation windows) but does not convey prediction uncertainty across the 10 independent runs. This limits the reader's ability to assess:

•        Model reliability at different temporal scales

•        Whether certain aggregation intervals show higher prediction variance

**Summary:**

The manuscript is recommended for publication if the above suggestions are addressed or answered.

---

## Author Comment (AC5)

**Sam Houston State University**

**DEPARTMENT OF ENVIRONMENTAL AND GEOSCIENCES**

**Manuscript Number: egusphere-2025-3540**

Renjie Zhou and Shiqi Liu. "A hybrid Kolmogorov-Arnold networks-based model with attention for predicting Arctic River streamflow"

**Reviewer 1**:

Zhou and Liu present a novel approach for a data-driven model for discharge modelling. It is based on a Kolmogorov-Arnold network combined with a Long-Short Term Memory (LSTM) model, an attention mechanism that includes a trigonometric depiction of seasonal patterns, as well as a physics-based constrain. The newly developed model aimed at improving the prediction of discharge within arctic areas with their special characteristics like perma frost and accumulation and melting of snow over longer periods. Therefore, the model was applied to the discharge data of the Kolyma River in Siberia and the prediction evaluated against the predictions of several other simpler models.

I have found the presented modelling approach to be a novel and valuable contribution to the hydrological modelling community. I believe it to be fitting for the scope of the Journal. However, the presented manuscript needs work regarding the methodology section as well as the discussion.

Reply: We are grateful for the reviewer's positive feedback and constructive suggestions. We have thoroughly revised the manuscript, corrected errors, added references, addressed each comment, and provided the necessary clarifications as outlined below.

Major comments:

1. Line 30: I can't really support the statement that the presented framework is (better) suited for predicting Arctic River discharge under changing climate conditions. It is well likely that climate change impacts the respective catchments in a way that the general behaviour changes - which also alters how discharge forms. I then get to a model space where the model has to extrapolate - which data-driven models are unsuited for.

Reply: Implemented. We thank the reviewer for raising this point. We have revised the statement in the manuscript and added a short paragraph about its limitations. While the RCPIKLA model demonstrates robust performance for the Kolyma River prediction under historical and current hydroclimatic conditions, several limitations should be acknowledged. As a data-driven model trained on historical observations, the model's performance may degrade if climate change induces fundamental shifts in watershed behavior that extend beyond the range of training conditions. Such regime changes may include but are not limited to scenarios like transitions from continuous to discontinuous permafrost, and significantly altered seasonal patterns. Under such scenarios, the model would need to extrapolate beyond its training data range, which remains a challenge for data-driven approaches. Future applications under changing climate conditions should include regular model retraining and validation as new observations become available.

2. Line 137, Figure 1: I personally don't think the figure to be well chosen, as the important aspects are missing. I would rather use a fogure that shows the catchment itself with its topography.
   Reply: Implemented. We thank the reviewer for this advice. The figure has been updated to show the catchment itself with its topography. Also, the input variables over the entire time span will be plotted and provided along with the catchment map.

3. Line 138-143: These lines are unnecessary here and probably can be deleted. All those things have already been said within the introduction and are explained over the methodology section anyways.
   Reply: Implemented. These lines have been deleted and revised to increase the flow. The sections of introduction, study area, and data acquisition, and methodology are reorganized to improve the flow and reduce the overlap.

4. Line 144-145: All steps, that are necessary for actual model runs should come after the model description. Otherwise, the order is confusing.

   Reply: Implemented. It is reorganized to improve readability and clarity. The preprocessing step has been improved with the following introduction after the model description.

   Prior to model training, the input variables, including monthly precipitation, temperature and evapotranspiration data, are preprocessed and standardized using the Z-score normalization technique: $X_{std} = \frac{X-\mu}{\sigma}$, where $\mu$ and $\sigma$ are the mean and standard deviation computed from the training dataset; $X$ and $X_{std}$ denote the input values before and after standardization, respectively. This standardization process ensures that features with different scales contribute appropriately to the training process and improves model convergence.

5. Line 146-164: The description of the whole model structure should be done after the individual parts are explained. Figure 2 also should be moved there.

   Reply: Implemented. We have moved the whole model struction description after introducing all individual components.

   In summary, this newly proposed hybrid model leverages the KAN component as a feature transformation layer to extract and learn complex nonlinear patterns from hydrological and meteorological datasets. The LSTM component captures short- and long-term dependencies and effectively simulates sequential patterns and discharge variability. To further refine temporal learning, the attention mechanism is introduced and integrated, which allows the proposed model to selectively emphasize historically significant time steps, particularly those driving major and seasonal hydrological transitions. An important innovation is the residual compensation structure, which explicitly addresses the challenges of predicting extreme discharge events. By learning systematic error patterns, the

residual structure can adjust simulations based on residual predictions and improve performance during high-variability scenarios. Unlike conventional data-driven models that completely ignore fundamental physical constraints, the newly developed model incorporates physics-informed loss functions. Additionally, the model employs seasonality-aware encoding using trigonometric transformations to recognize the cyclic nature of hydrological processes. This architecture is designed to provide an accurate and robust framework for forecasting river discharge in Arctic and permafrost-dominated environments.

6. I do recommend the inclusion of an additional efficiency measure like KGE, that is complementary to the other ones and also incorporates different aspects of the discharge like bias for example. Please also cite and mention, which version of the KGE you use then.

Reply: Implemented. We have added KGE' (2012) as an additional evaluation metrics.

The following introduction is added to the subsection of 3.6 Evaluation Metrics. Also, pictures, references and discussion are revised and updated accordingly.

In addition to NSE and RMSE, the Kling-Gupta Efficiency (KGE) is employed to provide a balanced assessment of model performance. The KGE metric was developed to address certain limitations of NSE, particularly its sensitivity to extreme values and the potential compensation of errors in mean, variance, and correlation (Gupta et al., 2009). Unlike other metrics, KGE explicitly decomposes model performance into three components: linear correlation, bias ratio, and variability ratio. In this study, the modified KGE is employed, which addresses issues with the original formulation's sensitivity to the magnitude of standard deviations (Kling et al., 2012). The modified KGE (KGE') is calculated as:

$$KGE' = 1 - \sqrt{\left(r_{kge} - 1\right)^2 + \left(\beta_{kge} - 1\right)^2 + \left(\gamma_{kge} - 1\right)^2},$$

where $r_{kge}$ refers to the linear correlation coefficient between observed and simulated discharge; $\beta_{kge}$ refers to the ratio of simulated mean to observed

mean; $\gamma_{kge}$ denotes the variability ratio. The KGE' ranges theoretically from -∞ to 1, with KGE' = 1 indicating perfect agreement between observations and predictions in terms of correlation, bias, and variability. A KGE' value of -0.41 represents the performance of using the mean flow as a predictor, serving as a natural benchmark below which model predictions are no better than simply using the long-term average (Knoben et al., 2019). In hydrological modeling applications, KGE' values above 0.75 are generally considered very good, values between 0.5 and 0.75 indicate satisfactory performance, and values below 0.5 suggest unsatisfactory model performance (Towner et al., 2019). The use of multiple complementary metrics (NSE, RMSE, and KGE') provides a comprehensive evaluation framework. While NSE emphasizes matching variance and is sensitive to peak flows, KGE' provides balanced assessment across correlation, bias, and variability. RMSE quantifies absolute error magnitude in original units, which is particularly important for operational applications. Together, these metrics enable thorough assessment of model performance across different aspects of discharge prediction, from overall pattern matching to peak flow accuracy.

The KGE' metric provides additional insights into model performance by decomposing errors into correlation, bias, and variability components. The RCPIKLA model achieves KGE' values ranging from 0.74 to 0.82 across all time steps. Similar to NSE, the RCPIKLA model reaches its peak KGE' performance of approximately 0.82 at the 9-month time step. The baseline models demonstrate modest KGE' performance, with values ranging from 0.64 to 0.73. A notable degradation in KGE' performance is observed at the 12-month time step, where the RCPIKLA value drops to approximately 0.74, falling below the 0.75 threshold. This decline likely reflects the challenges of maintaining balanced performance across all three KGE' components (correlation, bias, and variability) at very long forecasting horizons. At 12 months, accumulated prediction errors and the increased difficulty in capturing seasonal phase transitions may cause the

model's predictions to exhibit greater bias or variability mismatch compared to observations, despite maintaining reasonable correlation.

Reference:

Gupta, H. V., Kling, H., Yilmaz, K. K., and Martinez, G. F.: Decomposition of the mean squared error and NSE performance criteria: implications for improving hydrological modelling, J. Hydrol., 377, 80–91, https://doi.org/10.1016/j.jhydrol.2009.08.003, 2009.

Knoben, W. J. M., Freer, J. E., and Woods, R. A.: Technical note: inherent benchmark or not? Comparing nash–sutcliffe and kling–gupta efficiency scores, Hydrol. Earth Syst. Sci., 23, 4323–4331, https://doi.org/10.5194/hess-23-4323-2019, 2019.

Towner, J., Cloke, H. L., Zsoter, E., Flamig, Z., Hoch, J. M., Bazo, J., Coughlan De Perez, E., and Stephens, E. M.: Assessing the performance of global hydrological models for capturing peak river flows in the Amazon basin, Hydrol. Earth Syst. Sci., 23, 3057–3080, https://doi.org/10.5194/hess-23-3057-2019, 2019.

7. Why does the methodology end here? Important parts that come up later within the results part are missing. The methodology should explain that the final model is compared to certain baseline models and how they distinguish from the new model presented here. Furthermore, the whole part is missing about how the model is trained on the data, with how many runs, ending criterion, hyper parameters and so on.

Reply: Implemented. We have restructured amd revised the manuscript and have add a subsection (Section 3.7) of model implementation and training to introduce the models and model differences, such as with how many runs, ending criterion, hyperparameters and so on.

**3.7 Model implementation and training**

As shown in Fig. 4, prior to model training, the input variables, including monthly

precipitation, temperature and evapotranspiration data, are preprocessed and standardized using the Z-score normalization technique: $X_{std} = \frac{X-\mu}{\sigma}$, where $\mu$ and $\sigma$ are the mean and standard deviation computed from the training dataset; $X$ and $X_{std}$ denote the input values before and after standardization, respectively. This standardization process ensures that features with different scales contribute appropriately to the training process and improves model convergence (LeCun et al., 1998).

In regions dominated by permafrost, snow accumulation and melt typically exhibit strong seasonal periodicity (Andersson et al., 2021; Ernakovich et al., 2014). Discharge patterns are strongly influenced by annual cycles of temperature, snow accumulation, and melt in Arctic hydrological systems (Häkkinen and Mellor, 1992). Accurately capturing such periodic behaviors can help develop robust long-term forecasting models. To include these cyclical patterns and facilitate smooth temporal transition, a trigonometric encoding (TE) of seasonal features is incorporated as input variables using sine and cosine transformations of the calendar month. Specifically, the timestamp is encoded to two features using the following trigonometric transformations:

$$\text{Month}_{\sin} = \sin\left(2\pi \frac{m}{12}\right); \ \text{Month}_{\cos} = \cos\left(2\pi \frac{m}{12}\right),$$

where m refers to the calendar month m $\in$ {1,2,$\cdots$,12}. These encodings aim at capturing cyclical temporal patterns without introducing artificial discontinuities between December and January. The trigonometric features are concatenated with other input variables, including temperature, precipitation and evapotranspiration, and fed into the residual-compensated physics-informed KAN-LSTM model with attention.

Table 1 summarizes the hyperparameters and configuration settings used in this study. The choice of hyperparameters balances model capacity with overfitting risk, given the limited training data available. The LSTM hidden dimension of 64 units and a dropout rate of 0.3 prevent overfitting while capturing essential temporal patterns. The batch size and epoch size are set to 32 and 150,

respectively. The optimal physics constraint weight ($\beta = 0.3$) and the MSE weight ($\alpha = 0.7$) are adopted by conducting grid search over $\alpha \in \{0.1, 0.3, 0.5, 0.7, 0.9\}$ (Figure S1 in Supplementary Material). With these hyperparameters, the newly proposed model trained in the training dataset of the Kolyma River, and then the fine-tuned models are applied to the unseen testing dataset for the assessment of the predictive performance. The prediction performance is compared with several popular temporal baseline models, including the simple RNN, LSTM, and GRU models. To assess model stability and minimize the effects of stochastic processes in the training procedure, each model configuration is trained 10 times independently on Google Colab. This repeated training protocol allows assessment of performance variability arising from the inherent stochasticity in the optimization process, including random batch shuffling and numerical precision variations.

Table 1 Model hyperparameters and configuration settings

| Parameters | Values |
|---|---|
| Training Epochs | 150 |
| Batch size | 32 |
| Learning rate | 0.0005 |
| Optimizer | Adam |
| Early stopping patience | 10 |
| MSE weight ($\alpha$) | 0.7 |
| Physics constraint weight ($\beta$) | 0.3 |
| KAN grid size | 5 |
| KAN number of layers | 2 |
| LSTM hidden dim | 64 |
| Baseline models hidden dim | 64 |
| Dropout | 0.3 |
| Attention activation | Tanh |
| Output activation | ReLU |
| Number of runs | 10 |

References:

LeCun, Y., Bottou, L., Orr, G. B., and Müller, K.-R.: Efficient BackProp, in:

Neural networks: tricks of the trade, vol. 1524, edited by: Orr, G. B. and Müller,

K.-R., Springer Berlin Heidelberg, Berlin, Heidelberg, 9–50, https://doi.org/10.1007/3-540-49430-8_2, 1998.

Andersson, T. R., Hosking, J. S., Pérez-Ortiz, M., Paige, B., Elliott, A., Russell, C., Law, S., Jones, D. C., Wilkinson, J., Phillips, T., Byrne, J., Tietsche, S., Sarojini, B. B., Blanchard-Wrigglesworth, E., Aksenov, Y., Downie, R., and Shuckburgh, E.: Seasonal arctic sea ice forecasting with probabilistic deep learning, Nat. Commun., 12, 5124, https://doi.org/10.1038/s41467-021-25257-4, 2021.

Ernakovich, J. G., Hopping, K. A., Berdanier, A. B., Simpson, R. T., Kachergis, E. J., Steltzer, H., and Wallenstein, M. D.: Predicted responses of arctic and alpine ecosystems to altered seasonality under climate change, Global Change Biol., 20, 3256–3269, https://doi.org/10.1111/gcb.12568, 2014.

Häkkinen, S. and Mellor, G. L.: Modeling the seasonal variability of a coupled arctic ice-ocean system, J. Geophys. Res.: Oceans, 97, 20285–20304, https://doi.org/10.1029/92JC02037, 1992.

8. Line 323-327: This is methodology and should not be within the results part - as it is missing within the methods section.
   Reply: Implemented. This part has been removed from the results section to the methodology section.

9. Line 328-329: As mentioned earlier, the baseline models cannot be newly introduced within the results.
   Reply: Impelemented. We have changed the order of introduction. The baseline models are introduced in the methodology section before the resuls.

10. Line 343-344: You can't conduct boxplots. Do you mean you conducted the model application 10 times?
    Reply: Implemented. We have rephrased the manuscript to improve its clarity and

readability here We trained and evaluated each model in 10 independent runs. This repeated training quantifies performance variability due to the inherent stochasticity of the optimization process. Results from the 10 runs are summarized using boxplots.

11. Line 357: Figure 6 y axis seems to be cut, the numbers are partly missing
    Reply: Implemented. We thank the reviewer for pointing this out. The figures have been fixed.

12. Line 361: I dont see how this represents the "spectrum of hydrological variability". From my understanding, it is more of a possibility to see, how the model performs if the data is only available in lesser resolution. How does this assess the depiction of the hydrological variability?
    Reply: Implemented. We thank the reviewer for this important clarification. The reviewer is correct that our analysis examines model performance under varying flow conditions, from low to high discharge events. The corresponding description is rephrased for clarification.

13. Line 405: Figure 8, are these for a aggregation period of 1 month?
    Reply: We thank the reviewer for requesting this clarification. The boxplots in Figure 8 show results aggregated across all forecasting time steps. Each model variant is trained 10 times independently at each time step (1-12 months), yielding 120 total evaluations per model. The results of all 120 evaluations for each model are summerized in the boxplots. The manuscript has been revised for clarification.

14. Line 407-415: This is all methodology and not results.
    Reply: Implemented. We thank the reviewer for identifying this issue. The contents have been reorganized and moved to methodology.

15. Line 437-448: I dont think this part is really necessary here. The conclusion is not

a whole summary of the paper, but points out the key findings again.

Reply: Impelemened. This long paragraph has been removed.

16. Line 455-456: The river discharge has a long memory? The sentence does not make sense. I feel like there is a more thorough discussion necessary of why the model shows this behaviour regarding the model efficiency for different aggregation periods - where the reason must be within model structure and how it fits the discharge pattern over time.

Reply: Implemented. The sencence has been rephrased to avoid confusion. A more thorough discussion will be added here.

The optimal performance at the 9-month input sequence length reflects important temporal characteristics of this permafrost-dominated watershed and the model's capacity to capture structured temporal dependencies. In the Kolyma River basin, current discharge is influenced by hydrometeorological conditions that could span multiple seasons, such as snow accumulation, snowmelt dynamics, and subsequent baseflow recession controlled by active layer storage and permafrost-restricted groundwater flows. The 9-month optimal input window captures the information of seasonal dynamics which provides the model with sufficient temporal context. The attention mechanism further refines this by assigning higher importance to specific antecedent months that strongly influence current discharge. Shorter sequences may fail to capture full seasonal cycles and snow accumulation processes, while longer sequences (10-12 months) likely introduce temporal uncertainties.

17. I generally feel like the discussion part is lacking depth. While I personally recommend to separate results and discussion, you can keep both together if it makes sense overall. But in the current state, the results lack depth regarding the explanation of observed model behaviour. For example, line 462-463: has this been the same for the application of other models? Is this a common problem? Like this, a few more citations and comparisons to other studies would help

**Sam Houston State University**
**DEPARTMENT OF ENVIRONMENTAL AND GEOSCIENCES**

putting the paper within a broader context.

Reply: Implememted. We will improve the results and discussion.

This systematic underestimation of peak flows represents a common challenge in data-driven hydrological modeling, particularly for Arctic river systems, where extreme discharge events are relatively rare but carry significant implications for water resource management and hazard mitigation. Kratzert et al. (2019) observed similar patterns in LSTM-based rainfall-runoff modeling across diverse catchments. For Arctic rivers specifically, Gelfan et al. (2017) and Chang et al. (2025) reported that process-based models and machine learning approaches struggle with extreme conditions due to the complex processes and events that are poorly represented in limited observational records. In our study, extreme high discharge events (>80 mm) constitute less than 5% of the training dataset, creating a class imbalance problem common in hydrological time series (Nearing et al., 2021). The squared error loss function (MSE) used in model training inherently weights all samples equally, which can lead to optimization that favors the more numerous moderate flow events at the expense of rare extremes. Future work could address this limitation through specialized sampling techniques or physics-informed constraints specifically designed to better captures high-magnitude discharge events.

Reference:

Kratzert, F., Klotz, D., Shalev, G., Klambauer, G., Hochreiter, S., and Nearing, G.: Towards learning universal, regional, and local hydrological behaviors via machine learning applied to large-sample datasets, Hydrol. Earth Syst. Sci., 23, 5089–5110, https://doi.org/10.5194/hess-23-5089-2019, 2019.

Gelfan, A., Gustafsson, D., Motovilov, Y., Arheimer, B., Kalugin, A., Krylenko, I., and Lavrenov, A.: Climate change impact on the water regime of two great arctic rivers: modeling and uncertainty issues, Clim. Change, 141, 499–515, https://doi.org/10.1007/s10584-016-1710-5, 2017.

Chang, S. Y., Schwenk, J., and Solander, K. C.: Deep learning advances arctic

river water temperature predictions, Water Resour. Res., 61, e2024WR039053, https://doi.org/10.1029/2024WR039053, 2025.

Nearing, G. S., Kratzert, F., Sampson, A. K., Pelissier, C. S., Klotz, D., Frame, J. M., Prieto, C., and Gupta, H. V.: What role does hydrological science play in the age of machine learning?, Water Resour. Res., 57, e2020WR028091, https://doi.org/10.1029/2020WR028091, 2021.

18. Also, I am currently missing a graphical depiction of the gauging curve and the simulated discharge. I believe a figure for that would help to give the reader an idea of how the model behaves, where it might deviate from gauging data and where it is strongly in congruence with it.

    Reply: Implememted. A new graphic depiction of observed and simulated discharge will be added to the manuscript to provide the readers with a better idea of how different models behave.

Minor comments:

19. Line 22: structure

    Reply: Implemented. We have corrected the spelling/grammar error.

20. Line 24: dominated by permafrost

    Reply: Implemented. We have corrected the spelling/grammar error.

21. Line 27: ...that these components improve the predictive performance.

    Reply: Implemented. We have corrected the spelling/grammar error.

22. Line 46: These temperature dependent transitions...?

    Reply: Implemented. We have corrected the spelling/grammar error.

23. Line 128-129: Why is there no citation for the Dataset?

    Reply: Implemented. The data source and citation have been added to the manuscript.

24. Line 178: 1) Input expansion

    Reply: Implemented. We have corrected the spelling/grammar error.

25. Line 183-185: Kolmogorov-Arnold theorem while avoiding the computational overhead

   Reply: Implemented. We have corrected the spelling/grammar error.

26. Line 195: GELU

   Reply: Implemented. We have corrected the spelling/grammar error.

27. Line 196: Figure 3 not referenced within the text.

   Reply: Implemented. We have added Figure 3 in the text.

28. Line 200: ...mechanism and a hidden state, an LSTM can efficiently regulate...

   Reply: Implemented. We have corrected the spelling/grammar error.

29. Line 209: The memory cell of an LSTM is primarily composed...

   Reply: Implemented. We have corrected the spelling/grammar error.

30. Line 240: "Q refers the discharge prediction using the context vector calculated from the context vector." It has to be "refers to" and what is "using the context vector calculated from the context vector" supposed to mean?

   Reply: Implemented. It has been rephrased to improve clarity. We have corrected the spelling/grammar error.

31. Line 273: I recommend a semicolon after water.

   Reply: Implemented. We have corrected the spelling/grammar error.

32. Line 279: caused by sources, such as model simplifications...

   Reply: Implemented. We have corrected the spelling/grammar error.

33. Line 285-286: Maybe its better to reformulate the sentence and describe alpha and beta as parameters that have to be fitted through model application?

   Reply: Implemented. $\alpha$ and $\beta$ are weighting coefficients that control the relative importance of the data-driven loss (MSE) and physics-informed constraint terms in the combined loss function. The optimal physics constraint weight ($\beta = 0.3$) and the MSE weight ($\alpha = 0.7$) are adopted by conducting grid search over $\alpha \in \{0.1, 0.3, 0.5, 0.7, 0.9\}$.

34. Line 299: beneficial

   Reply: Implemented. We have corrected the spelling/grammar error.

35. Line 303-304: What is cited here? The Nash-Sutcliffe efficiency measure should

be properly cited.

Reply: Implemented. A reference has been added regarding the Nash-Sutcliffe efficiency measure.

36. Line 330: I would recommend to implement the name RCPIKLA of the new model earlier, instead of within the results.

Reply: Implemented. We have move it earlier.

37. Line 396: change "better captures"

Reply: Implemented. We have corrected the spelling/grammar error.

Sincerely yours,

*Renjie Zhou*

Renjie Zhou
Associate Professor
Department of Environmental and Geosciences
Sam Houston State University
Huntsville, TX 77340

---

## Author Comment (AC6)

**Sam Houston State University**

**DEPARTMENT OF ENVIRONMENTAL AND GEOSCIENCES**

**Manuscript Number: egusphere-2025-3540**

Renjie Zhou and Shiqi Liu. "A hybrid Kolmogorov-Arnold networks-based model with attention for predicting Arctic River streamflow"

**Reviewer 2:**

Summary: The manuscript is recommended for publication if the above suggestions are addressed or answered.

Reply: We sincerely thank the anonymous reviewers for the positive comments and constructive feedback. We have carefully revised the manuscript, added references and addressed comments in the manuscript.

1. One of the primary advantages of Kolmogorov-Arnold Networks is their enhanced interpretability compared to traditional MLPs. KAN is usually used to improve the interpretability of the relations between inputs and output, but there is no mention of that.

The manuscript fails to leverage or discuss this fundamental strength of KAN architecture. Specifically, there is no:

- Visualization of the learned univariate functions
- Symbolic regression analysis
- Interpretation of what relationships the KAN component discovered between hydrometeorological inputs and Arctic discharge
- Physical insights into the processes governing snowmelt-driven streamflow in permafrost regions

Include a dedicated subsection on KAN interpretability analysis containing:

- Visualization of learned activation functions for key input-output relationships
- Symbolic approximations of these functions where feasible (using symbolic regression tools available in KAN libraries)
- Physical interpretation of discovered patterns in the context of Arctic hydrology
- Comparison with known physical relationships in snowmelt hydrology from the

literature

Reply: Implemented. We thank the reviewer for this insightful comment regarding the interpretability advantages of Kolmogorov-Arnold Networks. We will add a new subsection to visualization of the learned univariate functions with symbolic regression analysis and discuss the interpretability of KAN.

**4.3. Interpretability analysis of Kolmogorov-Arnold Networks**

Kolmogorov-Arnold Networks can learn interpretable univariate functions that can be visualized and approximated symbolically (Liu et al., 2024). The learned activation functions from the KAN component for each input feature are derived and presented to examine how each hydroclimatic input is transformed prior to temporal aggregation by the LSTM-attention block. While the overall model remains a sequence model, the KAN component offers mechanistic insight into learned input transformations.

The learned univariate KAN functions for the primary hydroclimatic predictors and the seasonal encodings are plotted against standardized inputs. The learned mappings show distinct behaviors across variables. Temperature exhibits threshold-dependent behavior and an increasing response for positive standardized values, which are consistent with degree-day snowmelt formulations (Hock, 2003). The minimal response at very low temperatures reflects periods when all precipitation accumulates as snow with no melt contribution to discharge. The strengthening positive trend at high temperatures captures accelerated snowmelt during warmer periods and melt-season activation. The PET function remains relatively constant across most of the range but drops at extremely high PET values. This negative response at high evapotranspiration demand is physically meaningful in permafrost watersheds where shallow active layers and restricted groundwater storage make baseflow highly sensitive to evaporative losses during warm, dry periods. The transition may represent a threshold where evaporative water losses begin to substantially reduce streamflow, consistent with observations of increased Arctic river sensitivity to evapotranspiration under warming (Nijssen et al., 2001). Precipitation shows minimal direct transformation with a nearly flat or slightly negative function. It can be caused by winter precipitation accumulating as snow and contributing to discharge

only after spring melt, which creates multi-month lags (Gelfan et al., 2017). The learned functions for the temporal encoding variables ($Month_{sin}$ and $Month_{cos}$) shows how the KAN components represent seasonality. $Month_{sin}$ exhibits a clear, smoothly varying nonlinear transformation, whereas $Month_{cos}$ remains comparatively flat. The monotonic tendency in the $Month_{sin}$ curve suggests an asymmetric seasonal influence. It shows that the model responds differently to the rising and falling portions of the annual cycle, which is consistent with the sharp melt-season transition and the comparatively gradual recession that often follows peak flow. Importantly, because trigonometric encoding provides a continuous cyclical representation of annual timing, the KAN transformation can capture seasonal structure without introducing an artificial discontinuity at the year boundary.

It is worthwhile to note that, as a hybrid architecture, RCPIKLA is primarily interpretable at the KAN stage. As the KAN module represents input–feature mappings through learnable univariate functions, the learned curves and their symbolic approximations provide a transparent description of how each hydroclimatic predictor is transformed before being passed to the sequence model. However, this interpretability does not extend to a fully closed-form, end-to-end explanation of the final discharge prediction: the downstream LSTM block integrates information across multiple antecedent months and mixes transformed features through recurrent dynamics and temporal weighting. Consequently, the KAN-derived functions should be interpreted as input transformations, rather than as a complete mechanistic decomposition of the full temporal prediction process.

Reference:

Liu, Z., Wang, Y., Vaidya, S., Ruehle, F., Halverson, J., Soljačić, M., Hou, T. Y., and Tegmark, M.: KAN: kolmogorov-arnold networks, https://doi.org/10.48550/ARXIV.2404.19756, 2024.

Hock, R.: Temperature index melt modelling in mountain areas, J. Hydrol., 282, 104–115, https://doi.org/10.1016/S0022-1694(03)00257-9, 2003.

**Sam Houston State University**
**DEPARTMENT OF ENVIRONMENTAL AND GEOSCIENCES**

Nijssen, B., O'Donnell, G. M., Hamlet, A. F., and Lettenmaier, D. P.: Hydrologic sensitivity of global rivers to climate change, Clim. Change, 50, 143–175, https://doi.org/10.1023/A:1010616428763, 2001.

Gelfan, A., Gustafsson, D., Motovilov, Y., Arheimer, B., Kalugin, A., Krylenko, I., and Lavrenov, A.: Climate change impact on the water regime of two great arctic rivers: modeling and uncertainty issues, Clim. Change, 141, 499–515, https://doi.org/10.1007/s10584-016-1710-5, 2017.

2. what are the hyperparameters (epochs, batch size, learning rate) and details of the architecture of the RNN, GRU and other neural nets used for comparison.

The manuscript lacks essential details for all baseline models (RNN, GRU, LSTM):

- No specification of hyperparameters (epochs, batch size, learning rate)
- No architectural details (number of layers, hidden units, activation functions)
- No information about initialization methods
- No training procedure details (optimizer type, learning rate schedules, dropout rates)
- No stopping criteria or early stopping procedures
- No hardware specifications or training times

Reply: Implemented. A subsection of model implementation and training (Section 3.7) has been added to introduce the model architectures, hyperparamters, stopping criteria and other details.

**3.7 Model implementation and training**

As shown in Fig. 4, prior to model training, the input variables, including monthly precipitation, temperature and evapotranspiration data, are preprocessed and standardized using the Z-score normalization technique: $X_{std} = \frac{X-\mu}{\sigma}$, where $\mu$ and $\sigma$ are the mean and standard deviation computed from the training dataset; $X$ and $X_{std}$ denote the input values before and after standardization, respectively. This standardization process ensures

that features with different scales contribute appropriately to the training process and improves model convergence (LeCun et al., 1998).

In regions dominated by permafrost, snow accumulation and melt typically exhibit strong seasonal periodicity (Andersson et al., 2021; Ernakovich et al., 2014). Discharge patterns are strongly influenced by annual cycles of temperature, snow accumulation, and melt in Arctic hydrological systems (Häkkinen and Mellor, 1992). Accurately capturing such periodic behaviors can help develop robust long-term forecasting models. To include these cyclical patterns and facilitate smooth temporal transition, a trigonometric encoding (TE) of seasonal features is incorporated as input variables using sine and cosine transformations of the calendar month. Specifically, the timestamp is encoded to two features using the following trigonometric transformations:

$$\text{Month}_{\sin} = \sin\left(2\pi \frac{m}{12}\right); \ \text{Month}_{\cos} = \cos\left(2\pi \frac{m}{12}\right),$$

where m refers to the calendar month $m \in \{1, 2, \cdots, 12\}$. These encodings aim at capturing cyclical temporal patterns without introducing artificial discontinuities between December and January. The trigonometric features are concatenated with other input variables, including temperature, precipitation and evapotranspiration, and fed into the residual-compensated physics-informed KAN-LSTM model with attention.

Table 1 summarizes the hyperparameters and configuration settings used in this study. The choice of hyperparameters balances model capacity with overfitting risk, given the limited training data available. The LSTM hidden dimension of 64 units and a dropout rate of 0.3 prevent overfitting while capturing essential temporal patterns. The batch size and epoch size are set to 32 and 150, respectively. The optimal physics constraint weight ($\beta = 0.3$) and the MSE weight ($\alpha = 0.7$) are adopted by conducting grid search over $\alpha \in \{0.1, 0.3, 0.5, 0.7, 0.9\}$ (Figure S1 in Supplementary Material). With these hyperparameters, the newly proposed model trained in the training dataset of the Kolyma River, and then the fine-tuned models are applied to the unseen testing dataset for the assessment of the predictive performance. The prediction performance is compared with several popular temporal baseline models, including the simple RNN, LSTM, and GRU models. To assess model stability and minimize the effects of stochastic processes in the training

procedure, each model configuration is trained 10 times independently on Google Colab. This repeated training protocol allows assessment of performance variability arising from the inherent stochasticity in the optimization process, including random batch shuffling and numerical precision variations.

Table 1 Model hyperparameters and configuration settings

| Parameters | Values |
|---|---|
| Training Epochs | 150 |
| Batch size | 32 |
| Learning rate | 0.0005 |
| Optimizer | Adam |
| Early stopping patience | 10 |
| MSE weight ($\alpha$) | 0.7 |
| Physics constraint weight ($\beta$) | 0.3 |
| KAN grid size | 5 |
| KAN number of layers | 2 |
| LSTM hidden dim | 64 |
| Baseline models hidden dim | 64 |
| Dropout | 0.3 |
| Attention activation | Tanh |
| Output activation | ReLU |
| Number of runs | 10 |

In summary, this newly proposed hybrid model leverages the KAN component as a feature transformation layer to extract and learn complex nonlinear patterns from hydrological and meteorological datasets. The LSTM component captures short- and long-term dependencies and effectively simulates sequential patterns and discharge variability. To further refine temporal learning, the attention mechanism is introduced and integrated, which allows the proposed model to selectively emphasize historically significant time steps, particularly those driving major and seasonal hydrological transitions. An important innovation is the residual compensation structure, which explicitly addresses the challenges of predicting extreme discharge events. By learning systematic error patterns, the residual structure can adjust simulations based on residual predictions and improve performance during high-variability scenarios. Unlike conventional data-driven models that completely ignore fundamental physical constraints, the newly developed model incorporates physics-informed loss functions.

**Sam Houston State University**
**DEPARTMENT OF ENVIRONMENTAL AND GEOSCIENCES**

Additionally, the model employs seasonality-aware encoding using trigonometric transformations to recognize the cyclic nature of hydrological processes. This architecture is designed to provide an accurate and robust framework for forecasting river discharge in Arctic and permafrost-dominated environments.

References:

LeCun, Y., Bottou, L., Orr, G. B., and Müller, K.-R.: Efficient BackProp, in: Neural networks: tricks of the trade, vol. 1524, edited by: Orr, G. B. and Müller, K.-R., Springer Berlin Heidelberg, Berlin, Heidelberg, 9–50, https://doi.org/10.1007/3-540-49430-8_2, 1998.

Andersson, T. R., Hosking, J. S., Pérez-Ortiz, M., Paige, B., Elliott, A., Russell, C., Law, S., Jones, D. C., Wilkinson, J., Phillips, T., Byrne, J., Tietsche, S., Sarojini, B. B., Blanchard-Wrigglesworth, E., Aksenov, Y., Downie, R., and Shuckburgh, E.: Seasonal arctic sea ice forecasting with probabilistic deep learning, Nat. Commun., 12, 5124, https://doi.org/10.1038/s41467-021-25257-4, 2021.

Ernakovich, J. G., Hopping, K. A., Berdanier, A. B., Simpson, R. T., Kachergis, E. J., Steltzer, H., and Wallenstein, M. D.: Predicted responses of arctic and alpine ecosystems to altered seasonality under climate change, Global Change Biol., 20, 3256–3269, https://doi.org/10.1111/gcb.12568, 2014.

Häkkinen, S. and Mellor, G. L.: Modeling the seasonal variability of a coupled arctic ice-ocean system, J. Geophys. Res.: Oceans, 97, 20285–20304, https://doi.org/10.1029/92JC02037, 1992.

3. Recent papers suggest that KAN based architectures outperform classical ANN based architectures. There should have been a comparison with KAN based LSTM, GRU and other neural nets. The manuscript only compares RCPIKLA (which uses KAN) against traditional ANN-based models (RNN, GRU, LSTM), not against KAN-enhanced versions of these baseline architectures.

The comparison with no physics informed constraints and no residual has been

**Sam Houston State University**
**DEPARTMENT OF ENVIRONMENTAL AND GEOSCIENCES**

compared. However, the current experimental design still creates an attribution problem. Observed performance improvements could stem from:

- The KAN component specifically

- The attention mechanism

- The physics-informed constraints

- The residual compensation structure

- Seasonal trigonometric encoding

- Some synergistic combination of these components

Without proper ablation comparing LSTM-attention/KAN-LSTM/KAN-GRU versus RCPIKLA, the specific contribution of KAN remains unclear.

Reply: Implemented. We have added a new comparision model (KAN-enhanced baselime model) and a new evaluation metric and discussions to evaluate the contribution of KAN.

Also, we will analzye the inpretability of the KAN component.

The evaluation metrics of LSTM, KAN-LSTM (KAN transformation followed by LSTM without attention, physics constraints, or residual compensation), and RCPIKLA are compared and analyzed. We compoared the models with other baselines across multiple forecasting horizons (1-12 months), and plotted the distribution of metrics across 10 independent training runs. The comparison between LSTM and KAN-LSTM shows that KAN-based nonlinear feature transformation can produce consistent improvements across all time steps. Averaged across all forecasting horizons, KAN-LSTM achieves NSE of 0.77 ($\pm$0.025), RMSE of 9.4 mm ($\pm$0.68), and KGE' of 0.75 ($\pm$0.027), compared to LSTM's NSE of 0.70 ($\pm$0.034), RMSE of 10.94 mm ($\pm$0.61), and KGE' of 0.67 ($\pm$0.023). This represents approximately 12% improvement in NSE attributable specifically to KAN's learnable univariate functions. At the optimal 9-month time step, KAN-LSTM achieves NSE of 0.78 compared to LSTM's 0.70, which demonstrates that KAN provides substantial value for prediction.

4. The manuscript describes a physics-informed constraint that imposes an upper limit on predicted snowmelt contribution but does not explain the asymmetric treatment of constraint violations.

The asymmetric design requires clear physical justification:

- Upper bound rationale: Snowmelt contribution physically cannot exceed available snow water equivalent - this is a hard constraint based on mass conservation
- Lower bound question: Are underpredictions physically plausible? Could incomplete melting, refreezing, or sublimation make them valid? Or do they indicate model failure to capture melt processes?
- Bias implications: Does the asymmetric penalty introduce systematic bias toward underprediction?

Reply: Implemented. The asymmetric physics constraint used in the manuscript represents a simplification of complex Arctic hydrological processes with available data.

The snowmelt contribution calculated is one of the major contributors to the discharge rate in permafrost-dominated watersheds, such as the Kolyma River. While instantaneous discharge can legitimately fall below melt rates due to transient storage in the active layer, evapotranspiration losses, or refreezing during diurnal temperature fluctuations, these effects become negligible at the monthly aggregation scale in large, permafrost-dominated basins like the Kolyma River (Gusev et al., 2015). Continuous permafrost covering >90% of the Kolyma basin severely restricts subsurface infiltration and groundwater storage (Walvoord and Kurylyk, 2016; Woo et al., 2008). Unlike temperate watersheds where snowmelt can recharge deep aquifers, the impermeable permafrost layer forces meltwater to travel through the shallow active layer with limited storage capacity. Consequently, snowmelt rapidly converts to surface and near-surface runoff with minimal opportunity for long-term retention (Bring et al., 2016). Also, Arctic Rivers such as the Kolyma River and the Lena River exhibit strong discharge seasonality characteristic, with the majority of the annual discharge occurring during summer months (Ye et al., 2003). During these months, snowmelt represents the dominant water source,

**Sam Houston State University**
**DEPARTMENT OF ENVIRONMENTAL AND GEOSCIENCES**

and the monthly timestep aggregates over 30 days during which daily temperature fluctuations and local-scale heterogeneity in melt timing average out across the entire basin. While refreezing can occur during cold nights or sublimation during clear, windy days, these losses are small relative to the total melt flux at monthly basin-scale aggregation (Suzuki et al., 2015). Therefore, snowmelt represents a dominant and appropriate lower bound on discharge at this spatiotemporal scale. (Yang et al., 2002).

The asymmetric physical constraint in this study is designed and implemented to reflect both the availability of data and the scale-dependent hydrology of large permafrost-dominated Arctic watersheds. It is worthwhile to note that implementing symmetric upper bound constraints will further increase the physics-informed condition. Future studies should collect comprehensive data and develop more sophisticated, symmetric physics constraints that fully respect mass conservation while accounting for all water balance components.

Regarding bias implications, when pooling all residuals across horizons and runs, RCPIKLA obtains a low residual (0.08 mm, corresponding to +0.57% of the mean observed discharge), whereas RCKLA exhibits a negative mean residual (−0.31 mm, −2.23%). These results indicate that the physics-informed constraint does not introduce a systematic bias. Instead, it reduces the slight underprediction tendency of the unconstrained model and yields a more centered residual distribution overall.

References:

Gusev, E. M., Nasonova, O. N., and Dzhogan, L. Ya.: Physically based simulating long-term dynamics of diurnal variations of river runoff and snow water equivalent in the kolyma river basin, Water Resour., 42, 834–841, https://doi.org/10.1134/S0097807815060056, 2015.

Walvoord, M. A. and Kurylyk, B. L.: Hydrologic impacts of thawing permafrost—a review, Vadose Zone J., 15, 1–20, https://doi.org/10.2136/vzj2016.01.0010, 2016.

Woo, M.-K., Kane, D. L., Carey, S. K., and Yang, D.: Progress in permafrost hydrology

in the new millennium, Permafrost Periglacial Processes, 19, 237–254, https://doi.org/10.1002/ppp.613, 2008.

Bring, A., Fedorova, I., Dibike, Y., Hinzman, L., Mård, J., Mernild, S. H., Prowse, T., Semenova, O., Stuefer, S. L., and Woo, M. -K.: Arctic terrestrial hydrology: a synthesis of processes, regional effects, and research challenges, J. Geophys. Res.: Biogeosci., 121, 621–649, https://doi.org/10.1002/2015JG003131, 2016.

Ye, B., Yang, D., and Kane, D. L.: Changes in lena river streamflow hydrology: human impacts versus natural variations, Water Resour. Res., 39, 2003WR001991, https://doi.org/10.1029/2003WR001991, 2003.

Suzuki, K., Liston, G. E., and Matsuo, K.: Estimation of continental-basin-scale sublimation in the lena river basin, siberia, Adv. Meteorol., 2015, 1–14, https://doi.org/10.1155/2015/286206, 2015.

Yang, D., Kane, D. L., Hinzman, L. D., Zhang, X., Zhang, T., and Ye, H.: Siberian lena river hydrologic regime and recent change, J. Geophys. Res.: Atmos., 107, https://doi.org/10.1029/2002JD002542, 2002.

5. Physics-informed neural networks fundamentally rely on balancing multiple loss terms through weighting parameters. The manuscript mentions $\alpha$ and $\beta$ as weights for MSE loss and physics loss but does not report their values.

The manuscript must provide:

- Final $\alpha$ and $\beta$ values used for all reported results
- Scenarios of hit and trials
- Search space explored

Reply: Impelemnted. α and β are weighting coefficients that control the relative importance of the data-driven loss (MSE) and physics-informed constraint terms in the combined loss function. The optimal physics constraint weight ($\beta = 0.3$) and the MSE weight ($\alpha = 0.7$) are adopted by conducting grid search over $\alpha \in \{0.1, 0.3, 0.5, 0.7, 0.9\}$.

In addition, a new subsection is added to the Supplementary Material to introduce the search process and justify the optimal choise.

6. The manuscript lacks visualization of epoch-wise loss decomposition, which is important for assessment of convergence of all models. Without this analysis, it is impossible to assess whether the physics constraint meaningfully guides training or becomes negligible compared to the data-driven MSE loss.

Visualizing separate loss components reveals:

- Whether physics loss actually contributes to training or is overwhelmed by MSE loss
- Training stability and convergence behavior
- Potential issues: loss spikes, plateaus, phase transitions

Reply: We thank the reviewer for this suggestion to analyze training dynamics and loss component contributions. In the proposed model, a dual physics-guided approach with two components is implemented: 1) a snowpack layer, 2) a physics-informed loss constraint term. This manuscript included ablation analysis comparing models with and without both physics constraints. In Figure 10, it provided empirical validation and analzyed the results of RCPIKLA vs. RCKLA-no physics-informed components vs. PIKLA-no residual structure). We believe this analysis can address the reviewer's concern about the physics constraint's contribution.

7. Figure 6 (left): "y axis seems to be cut, the numbers are partly missing" - this affects readability and interpretation. Also, please check for spelling and grammatical errors throughout manuscript. Like a few spelling mistakes have been observed in abstract

Reply: Implemented. We thank the reviewer for pointing this out. We have carefully reviewed the manuscript to correct the spelling mistakes. The plots with missing y axis have been fixed and updated.

8. The physics-informed mechanism involves snow storage (S_t) and melt (M_t) terms that evolve over time. However, the manuscript does not specify:

- Initial values for S_0 and M_0 at the start of the simulation period
- How these initial conditions were integrated into the model?

Reply: We thank the reviewer for noting that the initial conditions were not explicitly stated. In our implementation, the snow storage (S_t) and melt (M_t) are initialized as zero at the beginning of each model input sequence for simplicity. Specifically, we set $S_0 = 0$ and $M_0 = 0$, and then update $S_t$ and $M_t$ recursively within the window based on precipitation and temperature. The computed term is integrated into the model by being added to the network-predicted discharge and by included in the physics-informed penalty term. In the future, a continuous state carryover across windows that maintains snow storage between consecutive sequences could be considered for future work.

9. It is mentioned conducting 10 independent runs but provides unclear or incomplete reporting of variability in results. Fig8 represents the rmse and nse RCPIKLA variants with all predictions, what is the average RMSE over 10 runs, how much variation is observed over independent runs?

Additionally:

- Figure 8 shows results (RMSE and NSE for RCPIKLA variants) but it's unclear whether these represent single runs, mean values, or distributions
- No explicit reporting of mean $\pm$ standard deviation for performance metrics
- No statistical significance testing comparing model variants

Reply: Implemented. We thank the reviewer for this important comment highlighting the need for comprehensive statistical reporting. We have substantially revised the figure and added statistical analysis to address the concerns. The figure has been recreated to be more informative. The figure caption is updated to explicitly state that each box plot

**Sam Houston State University**

**DEPARTMENT OF ENVIRONMENTAL AND GEOSCIENCES**

aggregates results across forecasting horizons (1-12 months) and independent training runs, which produces 120 data points per model (12 time steps).

Each model variant is trained 10 times independently at each time step (1-12 months), yielding 120 total evaluations per model. The performance metrics are aggregated and visualized in the boxplot. In Fig 10, it reveals that the complete RCPIKLA model achieves mean NSE of $0.827 \pm 0.030$ (mean $\pm$ standard deviation) across 120 evaluations, which represents significant improvements over the PIKLA model without residual compensation ($0.790 \pm 0.029$, $p < 0.001$) and the RCKLA without physics ($0.812 \pm 0.031$, $p < 0.001$). Similarly, RCPIKLA obtains lowest RMSE ($8.12 \pm 0.75$ mm) compared to PIKLA ($8.98 \pm 0.52$ mm, $p < 0.001$) and RCKLA ($8.47 \pm 0.76$ mm, $p < 0.001$). T-tests confirm performance differences are statistically significant at $p < 0.001$ level, which demonstrates that observed improvements are robust rather than artifacts of specific random initiations.

In summary, the ablation comparisons isolate individual component contributions: the residual structure (RCPIKLA vs PIKLA) improves NSE by 0.038 (4.8% relative improvement), while the physics-informed constraint (RCPIKLA vs RCKLA) contributes 0.015 NSE improvement (1.8% relative). Both components provide independent, statistically significant ($p < 0.001$) performance gains, confirming their complementary roles in the hybrid architecture. The synergistic integration of both components yields a new structure that balances data-driven flexibility with physical consistency. This hybrid approach is particularly advantageous in data-limited environments like Arctic Rivers, where the physics-informed constraints and the residual compensation help overcome model simplifications and data uncertainty.

10. Figure 5 currently shows model predictions at 12 time intervals (representing different aggregation windows) but does not convey prediction uncertainty across the 10 independent runs. This limits the reader's ability to assess:

- Model reliability at different temporal scales

**Sam Houston State University**
**DEPARTMENT OF ENVIRONMENTAL AND GEOSCIENCES**

- Whether certain aggregation intervals show higher prediction variance

Reply: Implemented. We thank the reviewer for this valuable suggestion to quantify prediction uncertainty across forecasting horizons. To make run-to-run uncertainty visible, we summarize and report model performance across 10 independent training runs for each aggregation window so that the readers can better evaluate model performance and relibatility at different temporal scales and prediction variances.

In the Supplementary Material, Tables S1–S3 report the mean, minimum, and maximum values values across runs for NSE, RMSE, and KGE' (2012) at each forecasting horizon. This should allow the readers to better evaluate model performance and relibatility at different temporal scales and prediction variances, without overcrowding the main figure with multiple model curves.

Sincerely yours,

*Renjie Zhou*

Renjie Zhou
Associate Professor
Department of Environmental and Geosciences
Sam Houston State University
Huntsville, TX 77340